# A high-speed search engine pLink 2 with systematic evaluation for proteome-scale identification of cross-linked peptides

Zhen-Lin Chen [1,2,4], Jia-Ming Meng [1,2,4], Yong Cao [3,4], Ji-Li Yin[1,2], Run-Qian Fang[1,2], Sheng-Bo Fan[1,2], Chao Liu[1,2], Wen-Feng Zeng[1,2], Yue-He Ding[3], Dan Tan[3], Long Wu[1,2], Wen-Jing Zhou[1,2], Hao Chi [1,2], Rui-Xiang Sun[3], Meng-Qiu Dong [3] & Si-Min He [1,2]

We describe pLink 2, a search engine with higher speed and reliability for proteome-scale identification of cross-linked peptides. With a two-stage open search strategy facilitated by fragment indexing, pLink 2 is ~40 times faster than pLink 1 and 3~10 times faster than Kojak. Furthermore, using simulated datasets, synthetic datasets, [15]N metabolically labeled datasets, and entrapment databases, four analysis methods were designed to evaluate the credibility of ten state-of-the-art search engines. This systematic evaluation shows that pLink 2 outperforms these methods in precision and sensitivity, especially at proteome scales. Lastly, reanalysis of four published proteome-scale cross-linking datasets with pLink 2 required only a fraction of the time used by pLink 1, with up to 27% more cross-linked residue pairs identified. pLink 2 is therefore an efficient and reliable tool for cross-linking mass spectrometry analysis, and the systematic evaluation methods described here will be useful for future software development.

[1] Key Laboratory of Intelligent Information Processing of Chinese Academy of Sciences (CAS), Institute of Computing Technology, CAS, Beijing 100190, China. [2] University of Chinese Academy of Sciences, Beijing 100049, China. [3] National Institute of Biological Sciences, Beijing 102206, China. [4]These authors contributed equally: Zhen-Lin Chen, Jia-Ming Meng, Yong Cao. Correspondence and requests for materials should be addressed to M.-Q.D. (email: dongmengqiu@nibs.ac.cn) or to S.-M.H. (email: smhe@ict.ac.cn)

Cross-linking mass spectrometry (CXMS) has emerged as an important tool for structural analysis of proteins and protein complexes[1–4]. It also has the potential to analyze protein–protein interaction networks at a proteome scale[5–7]. The idea of CXMS had long existed for structural interpretation of proteins, but its practice had been hindered by the lack of software tools. Over the past decade, many software tools have been developed to analyze CXMS data, such as xQuest[8], StavroX[9], pLink[10], xProphet[11], Protein Prospector[12], pLink-SS[13] (together with pLink referred to as pLink 1 hereinafter), Kojak[14], Xi[15], Xilmass[16], MetaMorpheusXL[17], and Xolik[18]. pLink 1 and xProphet were the first to propose a target-decoy based control of false discovery rate (FDR) and hence enabled CXMS to be applied to complex mixtures or even proteome-scale explorations in non-specialist laboratories[19]. Recent years have witnessed an explosion of successful applications of CXMS[20–24] and an increasing number of workflows have integrated CXMS to resolve the structures of protein complexes[25–28].

Although the CXMS approach has significantly improved, the increasing demand of interactome analysis—which involves identification of cross-linked peptides at a proteome scale—is challenging for CXMS software. Currently available software tools for CXMS data analysis suffer from poor speed and reliability. The first obstacle is a quadratically expanded search space, known as the $n$-square problem. Given $M$ spectra cross-linked among $N$ peptides, then matching $M$ spectra with possible peptide pairs of $O(N^2)$ may result in a time complexity of $O(MN^2)$ for an exhaustive search strategy, which is too high to support proteome-scale identification of cross-linked peptides. Later, the $n$-square problem was tackled by the open search strategy, which considers one cross-linked peptide pair as two single peptides, each bearing a modification of large mass yet unknown composition on linkable residues. This strategy identifies candidates for two single peptides individually and then recombine the top scored single peptides into cross-linked pairs based on the known mass of precursor[10,12–14,17]. In order to reduce the number of cross-linked pairs for fine-scoring, it usually filters single peptide candidates with a coarse-scoring and only the top-$k$ coarse-scored single peptides are kept and combined to find the best fine-scored peptide pair[10,12,14]. Thus, the coarse-scoring stage with time complexity $O(MN)$ becomes a new performance bottleneck in the open search strategy and further acceleration needs reduction of the number of coarse-scored peptides.

The second obstacle is the lack of systematic evaluation of credibility, i.e., precision and sensitivity of search engines. As the publications of pLink 1 and xProphet, the target-decoy approach (TDA) has become the principal method to control the FDR of cross-linked identifications and compare sensitivities. A number of studies further validated a search engine by mapping the cross-linked identifications to available crystallographic structures[8,10,12–14,16]. However, in solution, proteins are more dynamic and can exist in more conformations than reflected by crystal structures measured in condensed states, and hence the over-length cross-links may not be false[29]. Therefore, TDA becomes the only general method for credibility evaluation, but it has not been validated at a proteome scale, and whether or not to separately control the TDA–FDR for inter-protein and intra-protein cross-links is still controversial.

Here we present a search engine, pLink 2, to identify cross-linked peptides at a proteome scale with high efficiency and effectiveness. First, as we proposed earlier, a fragment index was introduced to reduce the number of coarse-scored peptides[30]. Different from the straightforward use of fragment index in MetaMorpheusXL[17], pLink 2 took full advantage of fragment index to reduce the number of coarse-scoring by a factor of 100, hence breaking the previous bottleneck of the open search strategy. In addition, based on the observation that, of the two linked peptides, one usually fragmented better than the other[12,15], pLink 2 adopted a two-stage open search strategy: for the first peptide, only the top-5 coarse-scored candidates were retrieved from the fragment index; then for each candidate of the first peptide, the mass of the second peptide can be deduced and used to retrieve all the candidates for the second peptide, whose number is usually small under high-accuracy mass spectrometry[31,32]. Speed comparison shows that pLink 2 was ~40 times faster than pLink 1 and 3~10 times faster than Kojak, at no cost of sensitivity.

Second, to evaluate the credibility of search engines for cross-linked peptide identification, we adapt ideas from related fields and introduce four TDA-independent methods including the use of simulated datasets[33–35], synthetic datasets[10,13], [15]N metabolically labeled datasets[36,37], and entrapment databases[38–40] to systematically evaluate the precision and sensitivity of pLink 2 and compare among ten different search engines. Each analysis method contained two different types of datasets cross-linked either by a chemical cross-linker (BS3 or Leiker[22]) or by disulfide bonds. We show that the proposed four TDA-independent evaluation methods are indispensable for systematic evaluation of CXMS search engines. After rigorous evaluation, it is clear that pLink 2 has achieved the highest precision and sensitivity by a large margin among the ten state-of-the-art tools.

Finally, to demonstrate the versatility and performance of pLink 2, we re-analyzed four proteome-scale cross-linking datasets of *Escherichia coli*, *Caenorhabditis elegans*, or human cells that had previously been analyzed using pLink 1[10,13]. Up to 27% more cross-linked residue pairs were identified only in a fraction of the time used before. These results show that pLink 2 is capable of supporting interactome analysis of higher eukaryotes by CXMS.

## Results

**Experimental design of systematic evaluation.** Evaluations were performed on four types of datasets in order of increasing difficulty, i.e., simulated datasets, synthetic datasets, [15]N metabolically labeled datasets, and entrapment databases. The simulated datasets were the simplest in that they were generated in idealized conditions with full annotations. Only the search engines that have achieved high precision and sensitivity on the simulated datasets will be selected for further evaluations using three types of real-world datasets. In the latter groups, the synthetic datasets were fully annotated, the [15]N metabolically labeled datasets were semi-annotated, whereas the datasets used in the entrapment database method had no ground truth or labeling information and hence were the most challenging. Detailed information for the 10 search engines used in this study is shown in Supplementary Table 1 and the detailed information of 12 datasets used in this study is shown in Supplementary Table 2.

**Workflow of pLink 2.** For a given cross-linked peptide pair, the α-peptide is defined as the one with the higher coarse-score, whereas the lower coarse-scored one is the β-peptide. In general, the α-peptide is often longer than β-peptide and is typically more informatively fragmented[12,15].

Cross-linked peptide pairs are identified using a two-stage open search strategy (Fig. 1a). First, MS1 scans are preprocessed by pParse, which extracts, after calibration, multiple precursor ions for each MS2 scan[41]. Next, α-peptides and β-peptides are retrieved in two stages. In the first stage, the α-peptide candidates are retrieved from the fragment index by query peaks that are generated from the MS2 spectrum (Fig. 1b). In the second

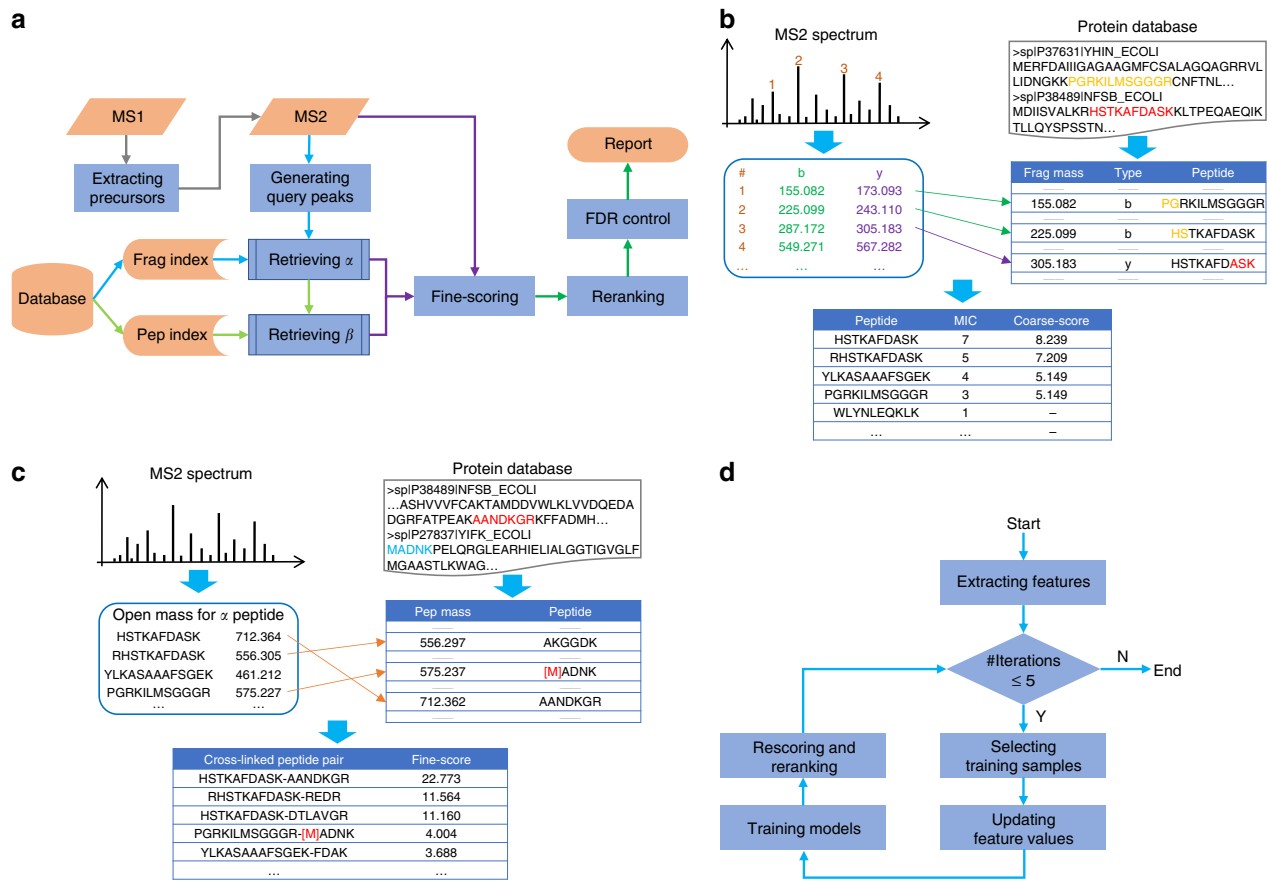

**Fig. 1** pLink 2 workflow. **a** The general workflow. Step 1, MS1 scans are preprocessed by pParse to extract precursor candidates. Step 2, for each MS2 spectrum, α-peptide candidates are retrieved from the fragment index using query peaks generated from the spectrum. Step 3, β-peptide candidates are retrieved from the peptide index using the complementary masses of α-peptides. Step 4, α- and β-peptide candidates are paired and fine-scored with the MS2 spectrum. Step 5, all top scored PSMs are re-ranked and filtered after FDR control. **b** The sub-workflow of α-peptide retrieval. For each MS2 spectrum, the peaks are converted into regular *b*, *y* ions to query the fragment index. Only those peptides with at least two matched ions are coarse-scored with the spectrum and the top-5 coarse-scored α-peptide candidates are kept. **c** The sub-workflow of β-peptide retrieval. For each α-peptide candidate, the open mass is first calculated by subtracting the α-peptide mass and the cross-linker mass from the precursor mass, and this mass is used to retrieve β-peptide candidates from the peptide index. Then, each of the five α-peptide candidates is paired with each of its complementary β-peptide candidates and these pairs are fine-scored with the spectrum. Finally, the highest fine-scored peptide pair is kept. **d** The re-ranking algorithm. PSMs are grouped into intra-protein, inter-protein, loop-linked, mono-linked, and regular groups, and a semi-supervised learning algorithm is used to re-score and re-rank them in each group

stage, the mass of β-peptide candidate, equal to the precursor mass minus the sum of the α-peptide candidate mass and the cross-linker mass, is used to retrieve the β-peptide candidates from the peptide index (Fig. 1c). Then, each α-peptide candidate is paired with each of the complementary β-peptide candidates and fine-scored against the MS2 spectrum.

Subsequently, pLink 2 searches for loop-linked, mono-linked, and regular peptides in the same way as pFind[42]. The highest scored non-cross-linked peptide or cross-linked peptide pair is kept for each spectrum after considering all possibilities. Then, the top scored candidate for each spectrum is separated into one of five groups: intra-protein, inter-protein, loop-linked, mono-linked, and regular peptide spectrum matches (PSMs). Each group of PSMs is re-ranked by a semi-supervised learning algorithm (Fig. 1d) that is similar to the widely used Percolator algorithm[43]. This separation ensures proper machine learning and FDR control, as the features and original error rates of each group differ from one another[11,12]. Finally, PSMs are filtered within group according to the specified thresholds (e.g., 5% FDR at PSM level).

**Credibility evaluation using simulated datasets.** As so many cross-linking search engines have been developed, tests on the simulated datasets were used as a qualification for further evaluations. Specifically, two rather simple simulated datasets were constructed: Simulated-BS3 and Simulated-SS, cross-linked by BS3 and disulfide bonds, respectively. Each dataset consists of 10,000 spectra from the cross-linking of 100 *E. coli* proteins (see Methods). The results of ten search engines on the Simulated-BS3 dataset are illustrated in Table 1, as sorted by sensitivity in ascending order.

The analysis showed that exhaustive search engines are not necessarily more sensitive. For xQuest, running light-only searches threw a division-by-zero exception. Xilmass threw an out-of-memory exception after 5 h of running with 32 GB memory. Xolik finished the search in 0.7 min, but the sensitivity was only 40.7%. This was in agreement with the test result on a HeLa cell dataset[18] (Supplementary Fig. 1). StavroX finished the search in 6 h and recalled 50.4% of cross-linked spectra at a precision of 78.6%, lower than the expected value considering that the FDR was set as 5%.

**Table 1 The performance of ten search engines on the Simulated-BS3 dataset[a]**

| Search engine | Search strategy | Sensitivity (%) | Precision (%) | Run time (Min) | Selected |
|---|---|---|---|---|---|
| xQuest[b] | Exhaustive | – | – | – | No |
| Xilmass[c] | Exhaustive | – | – | – | No |
| Xolik | Exhaustive | 40.7 | 93.7 | 0.7 | No |
| StavroX | Exhaustive | 50.4 | 78.6 | 363.9 | No |
| Xi | Open | 62.5 | 59.1 | 9.3 | No |
| MetaMorpheusXL | Open | 71.0 | 97.7 | 0.3 | No |
| Protein Prospector | Open | 78.6 | 97.2 | 16.9 | No |
| Kojak | Open | 85.3 | 97.8 | 1.7 | Yes |
| pLink 1 | Open | 99.8 | 99.8 | 12.3 | Yes |
| pLink 2 | Open | 99.9 | 100.0 | 0.5 | Yes |

[a] For sensitivity, precision, and run time, the average values obtained using three randomly generated Simulated-BS3 datasets were shown
[b] xQuest threw an exception "Illegal division by zero at /home/xqp/xquest/V2_1_1/xquest/bin/compare_peaks3.pl line 2246" and did not report any results
[c] Xilmass threw an exception "java.lang.OutOfMemoryError: GC overhead limit exceeded" and did not report any results

The open search engines all finished searching in a reasonable time. As Protein Prospector ran on a web server and did not provide an FDR value, we thus controlled FDR using the classical method[10]. At 5% FDR, pLink 2 obtained the highest sensitivity (99.9%) and precision (100.0%), followed in descending order by pLink 1, Kojak, Protein Prospector, MetaMorpheusXL, and Xi. Using the Simulated-SS dataset, the similar results were obtained (Supplementary Table 3).

As the simulated datasets were generated under idealized conditions and were rather simpler compared with the real-world data, all of the qualified search engines were expected to achieve high sensitivity and precision. However, only pLink 1 and pLink 2 reached above 95% in both aspects. Considering the sensitivity, precision, search speed, and usability, Kojak, pLink 1, and pLink 2 were selected for further evaluation.

**Credibility evaluation using synthetic datasets**. Next, we compared Kojak, pLink 1, and pLink 2 on two previously reported datasets of chemically cross-linked synthesized peptides, Synthetic-BS3[10] and Synthetic-SS[13], which were cross-linked by BS3 and disulfide bonds, respectively. The Synthetic-BS3 dataset was obtained from cross-linking experiments among 38 synthetic peptides, consisting of fully annotated 1030 positive PSMs and 1047 negative PSMs. The Synthetic-SS dataset was obtained from cross-linking experiments among 72 synthetic peptides, consisting of fully annotated 2289 positive PSMs and 2711 negative PSMs (see Methods).

Take the Synthetic-BS3 dataset as an example. We first searched all 2077 spectra against the smallest database containing only the sequences of 38 synthetic peptides (original database). A total of 904 spectra were correctly identified consistently by Kojak, pLink 1, and pLink 2, so these spectra were unbiased to three search engines (Fig. 2a). We therefore took these 904 PSMs as a new and fair standard dataset for searching against increasingly larger databases generated by appending the E. coli (4489 proteins, downloaded from Uniprot on 2017–10–23), worm (28,233 proteins, downloaded from WormBase on 2017–10–23), or human (71,579 proteins, downloaded from Uniprot on 2017–10–23) as an entrapment database to the original database. Supplementary Table 4 lists the search parameters.

As expected, for all three search engines, the numbers of correctly identified PSMs decreased as the database size increased (Fig. 2b). However, the sensitivity and precision of pLink 2 decreased only slightly; even with the huge human entrapment database, it recalled 97.0% PSMs while maintaining 98.7% precision (Supplementary Fig. 2). This is followed closely by pLink 1. In contrast, the sensitivity of Kojak decreased

significantly, down to ~57% with both the worm and the human entrapment databases.

In the open search stage of pLink 2, we recorded the ranks of correct α- and β-peptides among all candidates. As distraction gets worse as the entrapment databases become larger[44,45], the ranks of both correct α- and β-peptides decreased with the larger entrapment databases (Fig. 2c, d). However, thanks to sufficient fragmentation of α-peptides in most cases[12,15] and the well-designed fragment index (see Fragment index based α-peptide retrieval in Methods), the sensitivity of the coarse-scoring algorithm of pLink 2 was as high as 98.8%, i.e., it was able to retain the correct α-peptide sequence among the top-5 highest scoring candidate sequences for 98.8% of the time, even when searching against the human database (Fig. 2c). In contrast, as β-peptides do not fragment as well, their ranks decreased significantly as the entrapment database size increased (Fig. 2d). Looking into the pLink 2 search process, we found that only 84.0% of the correct β-peptides ranked within top-250, which is the default cutoff in Kojak, so it may be the reason why the sensitivity of Kojak decreased significantly when the worm and human databases were added to the search (Fig. 2b).

It is noteworthy that pLink 1 keeps the top-500 α-candidates and the top-500 β-candidates separately, whereas Kojak keeps both the α- and β-candidates together in the top-250; thus, the search space for fine-scoring with pLink 1 is much larger than that with Kojak[46]. As a result, although they have similar search strategies, pLink 1 performed much better than Kojak. pLink 2, in contrast, applies a two-stage open search strategy that is facilitated by both fragment indexing and peptide indexing. It not only retrieves correct α-peptides very efficiently, but also does not lose any β-peptide candidates. The results obtained for the Synthetic-SS dataset were similar to those obtained for the Synthetic-BS3 dataset (Supplementary Figs. 3 and 4). Collectively, the results from this performance evaluation using synthetic datasets showed that pLink 2 outperformed the other two search engines.

In addition to Kojak, pLink 1, and pLink 2, seven other search engines were also evaluated using the two synthetic datasets (Supplementary Note 1). The results show that pLink 2, pLink 1, and Kojak were indeed the top-3 highest-sensitivity search engines, in agreement with the conclusion drawn from the simulated datasets.

**Credibility evaluation using metabolically labeled datasets**. Then, two 1:1 $^{15}$N metabolically labeled datasets, E.coli-Leiker-$^{15}$N and E.coli-SS-$^{15}$N, were prepared and used to evaluate the precision of search engines using the percentage of PSMs with invalid quantitation ratios (Fig. 3a). Given the experimental

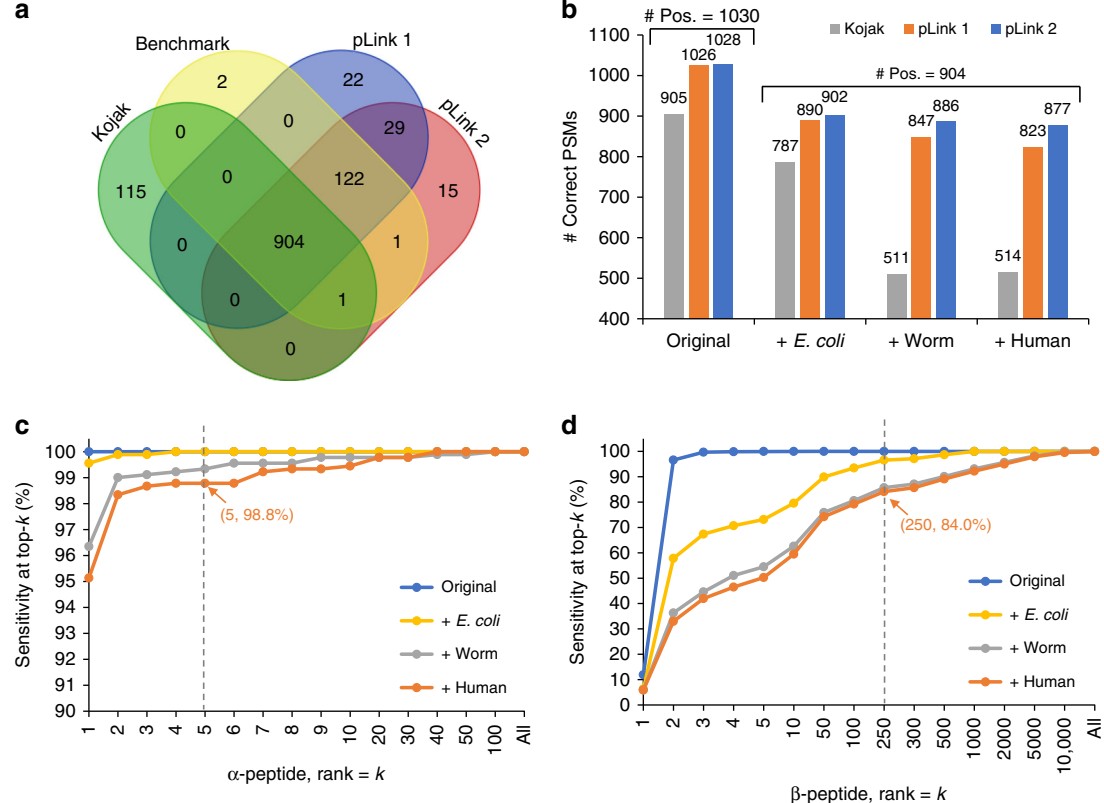

**Fig. 2** Performance evaluation on the Synthetic-BS3 dataset. **a** Venn diagram for the results of Kojak, pLink 1, pLink 2, and the benchmark. A total of 904 PSMs were correctly identified consistently by the three engines; these were used to be a new and fair standard dataset. **b** The numbers of correctly identified PSMs by each search engine. **c** The percentage of correct α-peptides ranking in the top-*k* in the open search stage of pLink 2. **d** Similar to **c**, but for β-peptides. The "Original" database contains only the sequences of 38 synthetic peptides, "+ *E. coli*" database contains sequences from the "Original" database and the *E. coli* whole proteome database, and "+ Worm" and "+ Human" are similar to "+ *E. coli*"

design, if a spectrum is identified as an unlabeled peptide or peptide pair, the corresponding $^{15}$N-labeled precursor ion should be observed in the MS1 scans. The mass distance between the unlabeled and $^{15}$N-labeled precursors is determined by the number of nitrogen atoms. In general, a falsely identified peptide or peptide pair carries an unexpected number of nitrogen atoms; thus, the chance is small that the calculated $^{15}$N-labeled version happens to have matched signals in the experimental spectra. As such, falsely identified peptides or peptide pairs will most likely have invalid quantification ratios, i.e., Not-a-Number (NaN), when quantified by pQuant[47] (Supplementary Fig. 5). The percentage of NaN ratios reflects the collective confidence level of given PSM identifications; a smaller percentage of NaN ratios corresponds to higher precision.

Take the E.coli-Leiker-$^{15}$N dataset as an example. To guarantee the performance of the $^{15}$N-labeling experiment, we first analyzed the unlabeled regular peptides (Supplementary Fig. 6a). A total of 45,393 regular PSMs were identified by pFind[37], of which only 0.3% were quantified as NaN ratios by pQuant[47]. In addition, the median of all valid quantification ratios was 1.03:1.00, showing that the performance of the $^{15}$N-labeling experiment was good. Then, three search engines were used to identify unlabeled cross-linked peptide pairs (Supplementary Table 5). At 5% FDR, pLink 2 reported the largest number of cross-linked PSMs (5196) with only 0.5% NaN ratios, whereas Kojak reported about half of that (2672) with 6.4% NaN ratios (Fig. 3b). pLink 1 came in between, reporting 4774 cross-linked PSMs with 3.8% NaN ratios.

With 5% FDR at the PSM level, pLink 1 reported the most cross-linked peptides (558), of which 5.4% were NaN-ratio peptides that were supported only by PSMs with NaN ratios

(Fig. 3c). Although pLink 2 reported slightly fewer cross-linked peptides (541), only 1.3% of them were NaN-ratio peptides. So, the number of credible (non-NaN ratio) peptides reported by pLink 2 (534) was actually larger than that reported by pLink 1 (528). Kojak reported the fewest cross-linked peptides (467), with 16.9% NaN ratios. Results uniquely identified by each search engine were also investigated; Supplementary Fig. 7 shows that the PSMs and peptides uniquely identified by pLink 2 had the lowest percentage of NaN ratios.

The percentage of NaN ratios is an independent criterion to evaluate precision and we can also use it to compare different TDA–FDR control strategies. Several studies have shown that the use of separate control of the FDR for inter-protein and intra-protein identifications, rather than global control, is an effective means of improving credibility of inter-protein results[11,12]. Our data analysis confirmed this and found that intra-protein PSMs and inter-protein PSMs show different changes when switching from global FDR control to separate FDR control (Fig. 3d, e). For intra-protein PSMs, more results were reported under separate FDR control and its percentage of NaN ratios was only slightly higher than that under global FDR control (Fig. 3d). For inter-protein PSMs, many fewer results were reported under separate FDR control and its percentage of NaN ratios decreased notably (Fig. 3e). This phenomenon also existed in the results of Protein Prospector (Supplementary Fig. 8), which recommended separate FDR control in its study[12]. Furthermore, our theoretical analysis about the relationship between the global FDR and subgroup FDRs of intra-protein and inter-protein identifications was accordant with the above experimental phenomenon; please see Supplementary Note 2 for details.

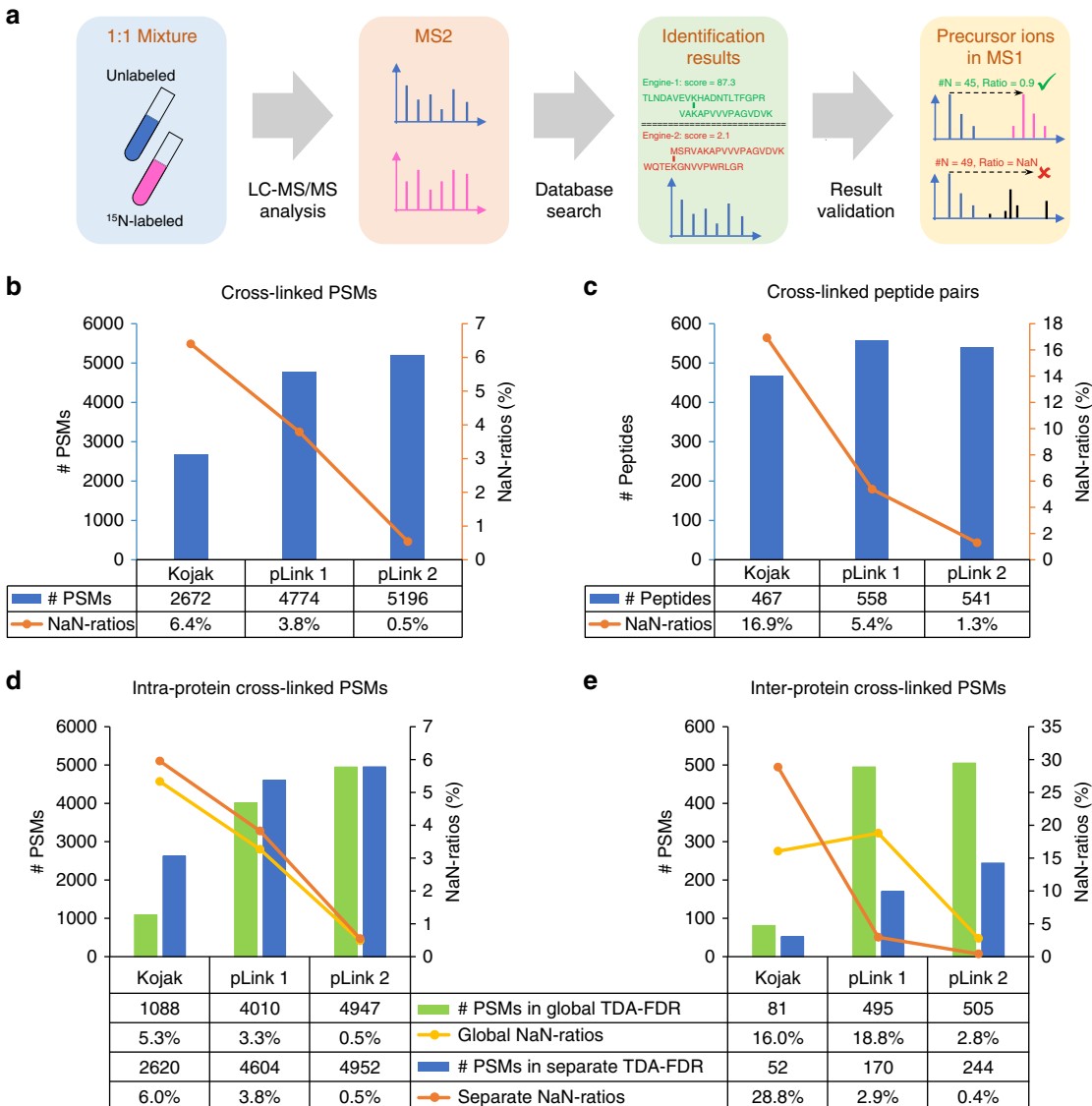

**Fig. 3** Performance evaluation on the E.coli-Leiker-15N dataset. **a** Experimental design of the E.coli-Leiker-15N dataset. The unlabeled and 15N metabolically labeled *E. coli* lysates were cross-linked separately, mixed at a 1:1 ratio, digested with trypsin, and analyzed by LC-MS/MS. The dataset was searched only for the unlabeled peptides using different search engines, and the identification results were passed to pQuant to quantify the intensity ratio of the 15N-labeled precursor to the unlabeled precursor. Lastly, the precision of identifications was investigated by checking the percentage of NaN-ratio PSMs and peptides. **b** Analyses of the identified cross-linked PSMs. **c** Analyses of the identified cross-linked peptide pairs. The histograms denote the total numbers of **b** PSMs or **c** peptide pairs identified by each search engine under separate FDR control of intra-protein and inter-protein results, and the curves denote the percentage of NaN-ratio **b** PSMs or **c** peptide pairs in the corresponding histograms. **d** For intra-protein PSMs, more results were reported under separate FDR control and its percentage of NaN ratios was slightly higher than that under global FDR control. **e** For inter-protein PSMs, many fewer results were reported under separate FDR control and its percentage of NaN ratios decreased, especially for pLink 1

An interesting observation was that although pLink 2 reported many more inter-protein PSMs under global FDR control, its subgroup percentage of NaN ratios only increased to 2.8%, significantly lower than that of either Kojak (16.0%) or pLink 1 (18.8%). Considering that many more inter-protein results will be reported without causing a big increase in the subgroup percentage of NaN ratios, pLink 2 provides global FDR control as an option. In the present paper, however, all search results were filtered using the separate FDR control strategy.

Using the E.coli-SS-15N dataset, we obtained similar test results and conclusions (Supplementary Figs. 6b, 9 and 10). Based on the percentage of NaN-ratio results, a new kind of FDR called NaN-FDR can be estimated, different from the normal TDA-FDR (Supplementary Note 3)[37]. The results in Supplementary Note 3

together show that the conclusions we reached based on the estimated NaN-FDRs were essentially the same as those reached based on the percentage of NaN ratios. Worth noting is that in the process of streamlining NaN-ratio analysis, pLink 2 has integrated pQuant[47], making quantitative CXMS a routine tool to reveal protein structural dynamics in solution.

**Credibility evaluation using entrapment databases**. Finally, we compared the three search engines on two real-world samples: the SCF(FBXL3) complex cross-linked by BS3 (used by Kojak[14]) and the Ca_v1.1 complex cross-linked by disulfide bonds (used by pLink 1[48]).

Take the SCF(FBXL3)-BS3 dataset as an example. In addition to three cross-linked proteins SKP1, CRY2, and FBXL3, several

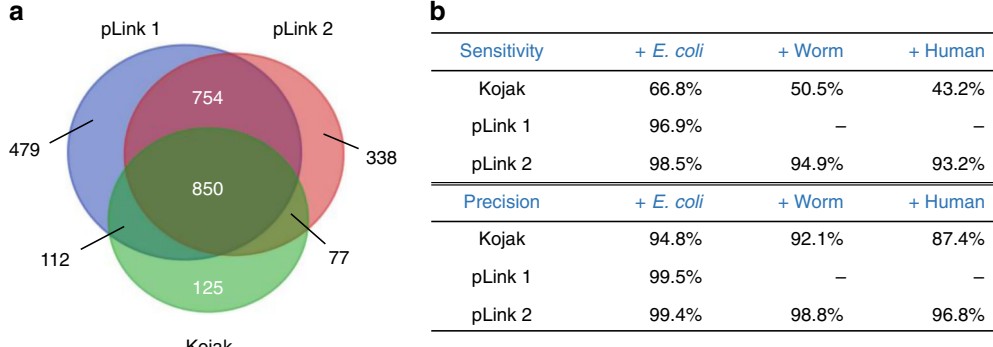

**Fig. 4** Performance evaluation on the SCF(FBXL3)-BS3 dataset. **a** A real-world protein complex sample was searched using Kojak, pLink 1, and pLink 2. A total of 850 cross-linked PSMs were identified consistently by the three engines; these were used to be a new and fair standard dataset. **b** The sensitivities and precisions of the three engines. "+ *E. coli*" database contains sequences from 146 target proteins and the *E. coli* whole proteome database and "+ Worm" and "+ Human" are similar to "+ *E. coli*". pLink 1 did not finish searching against the worm or the human entrapment databases within 1 week on a single computer when five variable modifications were set

*Drosophila melanogaster* proteins and lab contaminants were added to the sequence database, resulting in a total of 146 target proteins (original database)[14]. As we had done with the two synthetic datasets, the intersection of cross-linked PSMs identified by Kojak, pLink 1, and pLink 2 were then searched against the original database, to which increasingly larger entrapment databases (*E. coli*, worm, and human) were appended. An identification was deemed correct if it is identical to that in the intersection; otherwise it was considered incorrect. All parameters were the same as those in previously reported publications about Kojak[14] and pLink 1[13] (Supplementary Table 6).

Please note that the pLink 1 settings used in the present study yielded many more uniquely cross-linked residue pairs than did the pLink 1 settings used in the initial Kojak publication[14], where the precursor tolerance was set inappropriately (Supplementary Fig. 11). Although pLink 1 and pLink 2 reported many more PSMs than Kojak (Fig. 4a), the fact that this was a real-world sample made it difficult to assess and compare sensitivity and precision. Nevertheless, many studies have shown that the credibility of intersection results is greatly enhanced[49–51]. So, we took the 850 cross-linked PSMs consistently identified by the three search engines as a new and fair standard dataset to assess sensitivity and precision when searching against the original database plus entrapment databases of different sizes.

Similar to the results in the synthetic datasets, the sensitivity and precision of pLink 2 decreased only very slightly as the database size increased: it recalled 93.2% of the PSMs, while maintaining 96.8% precision, even with the huge human entrapment database (Fig. 4b). In contrast, the sensitivity and precision of Kojak decreased to 43.2% and 87.4%, respectively, with the human entrapment database. The performance of pLink 1 with the *E. coli* entrapment database was comparable to that of pLink 2, but unless a computer cluster is used, pLink 1 did not finish searching against the worm or the human entrapment databases within 1 week on a single computer when five variable modifications were set.

The results obtained for the Ca$_v$1.1-SS dataset were similar to those for the SCF(FBXL3)-BS3 dataset (Supplementary Fig. 12). This test once again illustrated that pLink 2 can maintain high sensitivity and precision even with a very large entrapment database, and hence is very useful when analyzing datasets at a proteome scale.

Among the four evaluation methods above, the simulated datasets and the synthetic datasets contain only mascot generic format (MGF) files. For search engines that do not accept MGF

files, we compared their performances using the $^{15}$N metabolically labeled datasets and the entrapment databases (Supplementary Notes 4 and 5).

**Speed evaluation of search engines.** Kojak, pLink 1, and pLink 2 were also compared in terms of computing time. Where possible, eight threads were used for each comparison, except when analyzing the Synthetic-BS3 and Synthetic-SS datasets with their original databases, because when fewer than 100 proteins are searched, pLink 1 takes an exhaustive approach that cannot support multi-threading.

The normalized computing times for the three search engines (Windows Server, Intel Xeon E5-2670 CPU with 32 cores, 2.6 GHz, 128 GB RAM) are shown in Table 2. On average, pLink 2 was 40 times faster than pLink 1 and was 3 times faster than Kojak. With the E.coli-SS-$^{15}$N dataset, pLink 1 took ~9 days and Kojak took ~14 h, whereas pLink 2 took only ~3 h. The speed-up on the Synthetic-SS dataset was obviously larger than the speed-up on the Synthetic-BS3 dataset. This difference was likely due to the fact that a majority of the precursor masses in the Synthetic-SS dataset are larger than those in the Synthetic-BS3 dataset (Supplementary Fig. 13), and both Kojak and pLink 1 examine all peptides whose masses are lower than the precursor mass in their coarse-scoring stages. Therefore, for Kojak and pLink 1, the higher the precursor mass, the larger the number of coarse-scored peptides, and the longer the search time. For pLink 2, thanks to both the fragment index and the MIC filter (Methods), fewer than 1% of candidates are retrieved and coarse-scored, resulting in a very efficient procedure for finding the top-$k$ coarse-scored single peptides (Supplementary Note 6). In addition, owing to the good fragmentation of α-peptides and the well-designed coarse-scoring algorithm, pLink 2 only kept the top-5 coarse-scored single peptide candidates during α-peptide retrieval, making the number of α–β combinations increase linearly with the number of peptides in the database. This time saving with pLink 2 is highly advantageous considering the intensive computing resources that are required when performing proteome-scale studies.

Beyond the fact that pLink 2 was ~3 times faster than Kojak, it was also notable that the sensitivity of pLink 2 was much higher than Kojak, even on the two synthetic datasets. Figure 2 and Supplementary Fig. 3 show that with pLink 2, ~16% of the correct β-peptides ranked beyond the top-250, the default value in Kojak. When more single peptides are kept by Kojak, its sensitivity

**Table 2 The normalized computing times of the three search engines on eight datasets**

| Dataset | pLink 1 | Kojak | pLink 2 | Real time[a] |
|---|---|---|---|---|
| Simulated-BS3 | 24.6 | 3.4 | 1.0 | 0.5 |
| Simulated-SS | 12.7 | 1.7 | 1.0 | 0.7 |
| Synthetic-BS3 + Original | 35.5 | 2.3 | 1.0 | 0.1 |
| Synthetic-BS3 + E. coli | 45.4 | 1.7 | 1.0 | 0.5 |
| Synthetic-BS3 + Worm | 50.8 | 2.4 | 1.0 | 3.5 |
| Synthetic-BS3 + Human | 38.3 | 2.1 | 1.0 | 6.4 |
| Synthetic-SS + Original | 32.7 | 2.6 | 1.0 | 0.2 |
| Synthetic-SS + E. coli | 60.2 | 3.3 | 1.0 | 1.2 |
| Synthetic-SS + Worm | 64.3 | 3.8 | 1.0 | 17.7 |
| Synthetic-SS + Human | 69.7 | 4.0 | 1.0 | 31.9 |
| E.coli-Leiker-$^{15}$N | 31.4 | 1.6 | 1.0 | 142.9 |
| E.coli-SS-$^{15}$N | 62.6 | 4.2 | 1.0 | 200.7 |
| SCF(FBXL3)-BS3 + Original | 20.0 | 1.7 | 1.0 | 260.6 |
| SCF(FBXL3)-BS3 + E. coli | 46.0 | 3.8 | 1.0 | 22.0 |
| SCF(FBXL3)-BS3 + Worm | –[b] | 4.5 | 1.0 | 275.7 |
| SCF(FBXL3)-BS3 + Human | –[b] | 4.2 | 1.0 | 385.1 |
| Ca$_v$1.1-SS + Original | 20.4 | 2.7 | 1.0 | 2.9 |
| Ca$_v$1.1-SS + E. coli | 31.8 | 3.4 | 1.0 | 1.8 |
| Ca$_v$1.1-SS + Worm | 36.1 | 4.0 | 1.0 | 34.5 |
| Ca$_v$1.1-SS + Human | 40.2 | 5.1 | 1.0 | 45.7 |
| Average | 40.2 | 3.1 | 1.0 | – |

[a]The real search times for pLink 2 are shown in minutes
[b]pLink 1 did not finish searching against the worm or the human entrapment databases within 1 week when five variable modifications were set

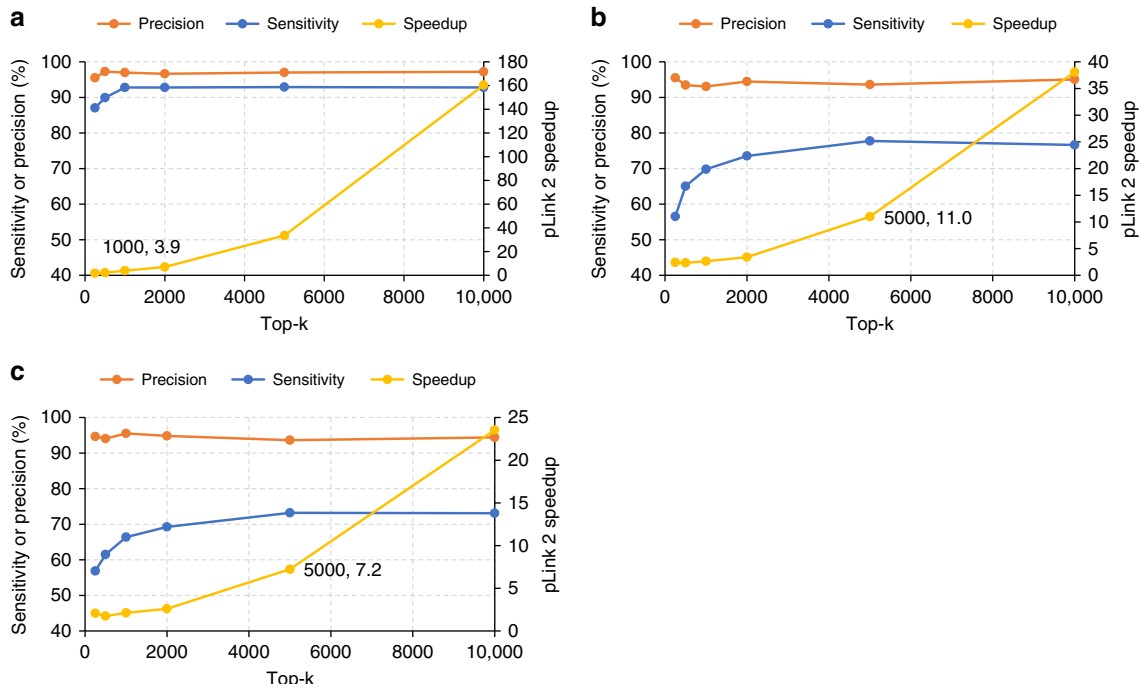

**Fig. 5** Increased speed of pLink 2 over Kojak on the Synthetic-BS3 dataset. **a** pLink 2 achieved a 3.9 times speed-up when searching against the *E. coli* entrapment database. The horizontal axis is the number of top-*k* scored single peptides kept in Kojak, starting from its default value of 250. Speed-up was measured when the sensitivity of Kojak remained steady. **b**, **c** Similar to **a**, but against **b** the worm and **c** the human entrapment database, respectively

increases and comes closer to that of pLink 2, and then comparing the running time would be more reasonable. Figure 5 shows that, upon increasing the number of single peptides kept by Kojak until its sensitivity remains steady, pLink 2 achieved a speed-up of 3.9, 11.0, and 7.2 for the *E. coli*, worm, and human entrapment databases, respectively, representing an average speed-up of 7.4 times over Kojak. This speed-up was 10.2 times on the Synthetic-SS dataset (Supplementary Fig. 14). An interesting observation is that even when the top-5000 scored single peptides were kept for the human entrapment database, the sensitivity of Kojak (73.2%) was still lower than that of pLink 2 (97.0%).

**Application of pLink 2 at a proteome scale**. To demonstrate the versatility and performance of pLink 2 at a proteome scale, pLink 2 was used to re-analyze four previously published datasets including *E. coli* and *C. elegans* whole-cell lysates cross-linked by Leiker, a BS3-like linker with enrichment function[22], and the *E. coli* and human disulfide proteomes[13] (Supplementary Note 7). In the original publications, PSM validations by TDA–FDR control were different for these two types of datasets, i.e., global FDR control for intra-protein and inter-protein PSMs in combination was applied to two Leiker datasets, separate FDR control was applied to two disulfide bond datasets, and 3121 (E.coli-Leiker), 882 (C.elegans-Leiker), 197 (E.coli-SS), and 553 (Human-SS) cross-linked residue pairs were identified by pLink 1, respectively.

In the present paper, pLink 2 applied the separate FDR control for all four datasets. A total of 2861 and 799 cross-linked residue pairs were identified for E.coli-Leiker and C.elegans-Leiker datasets, respectively, which were fewer in numbers but more credible than results of pLink 1 with the global FDR control. A total of 251 and 688 cross-linked residue pairs were identified for E.coli-SS and Human-SS datasets, respectively, which were 27% and 24% more than results of pLink 1, respectively. Supplementary Data 1−4 show detailed results for these four datasets.

Previously for pLink 1, large database search had been conducted on a computer cluster[10]. In the present study, all of the datasets were analyzed on a personal computer. For each of the cell lysate dataset, the search time was several months for pLink 1, but 1~3 days for pLink 2. These results highlighted that pLink 2 is capable of analyzing datasets cross-linked by a chemical cross-linker as well as native disulfide bonds, efficiently and effectively at a proteome scale.

## Discussion

CXMS is a valuable technique for investigating protein structures and protein–protein interactions, but data analysis at a proteome scale suffers from poor speed and credibility. Here we introduced pLink 2, which enables several substantial advances to the field of proteome-scale cross-linked peptide identification. First, we developed a two-stage open search strategy facilitated by fragment indexing. Benchmarking showed that pLink 2 was ~40 times faster than pLink 1 and 3~10 times faster than Kojak. Second, we designed four evaluation methods, using simulated datasets, synthetic datasets, [15]N metabolically labeled datasets, and entrapment databases, to systematically evaluate the credibility of search engines, including the precision and the sensitivity. Each evaluation method was performed on two different types of datasets cross-linked by a chemical cross-linker (BS3 or Leiker), and by disulfide bonds, respectively. pLink 2 achieved the highest precision and sensitivity among the ten state-of-the-art tools. More importantly, we demonstrated how to evaluate a search engine systematically, which is especially lacking in the field of CXMS. Third, we demonstrated the versatility and performance of pLink 2 on four previously published cell lysate datasets of E.coli-Leiker, C.elegans-Leiker, E.coli-SS, and Human-SS. pLink 2 took only a fraction of the time used by pLink 1, with up to 27% more cross-linked residue pairs identified.

Although pLink 2 is much faster compared with pLink 1 and Kojak, there is still room for improvement to achieve higher speeds. Sequence tags are short amino acid sequences that can be directly inferred from fragment peaks in MS2 and they have been used to screen peptide candidates in regular peptide search engines[37,52,53], e.g., a 5-tag has high specificity and can select only tens of peptide candidates for further scoring[37]. Supplementary Fig. 15 shows that 72%~87% of the identified cross-linked PSMs contained at least a 5-tag. That means, the number of coarse-scored peptides per spectrum is expected to be reduced from hundreds by fragment indexing to only tens by tag indexing, which will further speed up the cross-linked peptide identification.

Another way to overcome the *n*-square problem is through the use of MS-cleavable cross-linkers, which were not discussed in this paper. With MS-cleavable cross-linkers, such as PIR[54], BuUrBu[55], and DSSO[56], two peptides of one cross-linked peptide pair are detached, and can be identified by regular peptide search engines, making the cross-linked peptide identification in a linear time complexity[57]. Nevertheless, the fragment indexing used in pLink 2 can also be used to accelerate search engines for MS-cleavable cross-linked identification and systematic evaluations such as those proposed in this paper are still needed.

For CXMS, problems related to the credible identification still remain serious. On the one hand, the identification rate is still low. Supplementary Fig. 16 shows that the identification rates of four cell lysates were 35%~54%, indicating that about 50% of spectra were not interpreted. In contrast, the identification rate of spectra for regular peptide search can reach 70–85%[37]. Another similar challenge is the low abundance of cross-linked peptides in complex mixtures. Supplementary Fig. 16 also shows that even with the enrichable cross-linker Leiker[22], the identified cross-linked scans accounted for at most 15% of all identified scans and this proportion was no more than 5% on disulfide bond datasets, hence making a strong appeal for a more powerful enrichment method, either in sample preparation[22,58] or in data acquisition[15], or both.

On the other hand, the FDR controls at the peptide pair and residue pair levels still remain open. As correctly identified PSMs tend to cluster, while incorrectly identified PSMs tend to scatter and do not cluster to the same extent, FDR propagates from PSMs to peptide pairs and further to residue pairs[59]. The percentage of NaN ratios at PSM and peptide pair levels also showed similar propagation trends (Fig. 3).

## Methods

**Datasets**. A total of 12 datasets were used to evaluate the credibility of pLink 2. They can be categorized into two classes. The first class contained eight datasets serving for demonstration of four evaluation methods. Specifically, they were Simulated-BS3 and Simulated-SS for the simulated dataset evaluation, Synthetic-BS3 and Synthetic-SS for the synthetic dataset evaluation, E.coli-Leiker-[15]N and E. coli-SS-[15]N for the [15]N metabolically labeled dataset evaluation, and SCF(FBXL3)-BS3 and Ca$_v$1.1-SS for the entrapment database evaluation. These datasets were used to systematically evaluate the performance of different search engines, i.e., the speed, sensitivity, and precision, independent of common evaluation methods such as the TDA–FDR method and crystal structures.

The second class contained four previously published proteome-scale datasets already analyzed by pLink 1, including E.coli-Leiker, C.elegans-Leiker, E.coli-SS, and Human-SS, and they were re-analyzed to demonstrate the performance and versatility of pLink 2. The detailed information of these twelve datasets is shown in Supplementary Table 2.

**Preparation of the Simulated-BS3 and Simulated-SS datasets**. The fragmentation characteristics of synthetic peptides cross-linked by the BS3[10] and disulfide bonds[13] were used to generate Simulated-BS3 and Simulated-SS, respectively. The Simulated-BS3 dataset consists of cross-linked, loop-linked, mono-linked, and regular MS2 spectra, 2500 for each type, resulting in 10,000 MS2 spectra in total. As there are no mono-linked peptides in a disulfide bond sample, the Simulated-SS dataset only consists of 2500, 2500, and 5000 for cross-linked, loop-linked, and regular MS2 spectra, respectively, resulting in 10,000 MS2 spectra in total. The simulation method is described in Supplementary Note 8.

**Preparation of the Synthetic-BS3 dataset**. The Synthetic-BS3 dataset was a collection of 2077 annotated high-energy collisional dissociation (HCD) spectra obtained from 38 synthetic peptides cross-linked pairwise through BS3[10]. Of the 2077 spectra, 1030 were from light [d0]-BS3, and 1047 were from heavy [d4]-BS3. The 1047 spectra of cross-linked peptides containing heavy [d4]-BS3 served as negative samples, because in this study we searched with the mass of light [d0]-BS3 and the precursor mass tolerance was as small as ± 20 p.p.m.

**Preparation of the Synthetic-SS dataset**. The Synthetic-SS dataset was a collection of 5000 annotated HCD spectra obtained from 72 cysteine-containing synthetic peptides cross-linked pairwise through disulfide bonds[13]. Of the 5000 spectra, 2289 were high-quality HCD spectra of disulfide-linked peptide pairs obtained from 72 synthetic peptides and 2711 were negative samples that were not identified as regular peptides or disulfide-linked peptide pairs.

**Preparation of the E.coli-Leiker-$^{15}$N dataset**. Leiker bAL2[22] was used to cross-link the *E. coli* MG1655 whole-cell lysates. MG1655 was cultured in unlabeled and $^{15}$N-labeled M9 medium and collected at OD$_{600}$ 0.6–0.8. Pellets of 40 OD*ml *E. coli* cells was resuspended in 0.4 ml lysis buffer (50 mM HEPES, pH 7.5, 150 mM NaCl). Cell lysates were prepared using a FastPrep homogenizer (6.5 m/s, 20 s, repeat 5 times). After measuring protein concentration, 1 mg of unlabeled and $^{15}$N-labeled lysate was cross-linked separately with 0.33 mg [d0]-bAL2 for 0.5 h at room temperature. Cross-linking reactions were quenched with 20 mM ammonium bicarbonate. The two reactions were mixed together and then precipitated by trichloroacetic acid (TCA) followed by Trypsin digestion. After filtering with a 50 kDa cutoff Amicon Ultra-0.5 Centrifugal Filter Unit, the digested peptides were brought to a volume of 3 mL with 2% ACN, 20 mM HEPES, pH 8.2; the pH was adjusted to 10.0 with ammonia. High-pH reverse-phase separation was used for fractionation. The peptides were eluted with buffer B (80% ACN, 5 mM NH$_4$COOH, pH 10) gradient. A total of 39 two-minute fractions were collected and then combined into five fractions of similar shades of color (bAL2-linked peptides are bright yellow before cleavage of the biotin tag). Each pooled sample was evaporated to 200–300 µl before enrichment of bAL2-linked peptides on 50 µl high-capacity streptavidin beads. After release of beads and desalting through a C18 column, the five fractions of bAL2-linked peptides were analyzed by liquid chromatography–tandem mass spectrometry (LC-MS/MS) using a Q-Exactive HF mass spectrometer coupled with an EASY-nLC 1000 system (both from Thermo Fisher Scientific). Precursors of the $+ 1, + 2, + 8$, or above, or unassigned charge states were rejected and dynamic exclusion was set to 20 s.

**Preparation of the E.coli-SS-$^{15}$N dataset**. The *E. coli* strain BL21 was cultured in unlabeled and $^{15}$N-labeled NH$_4$Cl (Cambridge Isotope Laboratories, Inc.) M9 medium separately and collected at OD600 = 0.75. From a mixture of 35 OD*mL unlabeled and 35 OD*mL $^{15}$N-labeled cells, periplasmic proteins were prepared using the osmoticshock method[13]. Periplasmic proteins were released into 6 mL of pre-cooled solution of 5 mM MgSO$_4$ supplemented with 2 mM NEM and then precipitated on ice with 25% TCA followed by cold acetone wash twice. Precipitated proteins were air dried, resuspended in 8 M urea, 100 mM Tris, 2 mM NEM, pH 6.5. After the protein concentration was measured using the BCA Protein Assay Kit (Pierce), the sample was brought to 1–2 mg/mL, digested sequentially with Lys-C, trypsin, and Glu-C before SCX fractionation[60]. Eight SCX fractions were collected by sequential elution with 35 µl of 50, 150, 250, 350, 500, 650, and 800, and 1 M ammonium acetate, pH 2–3, at a flow rate of 1.0–2.0 µL/min. The LC-MS/MS analysis was performed on a Q-Exactive HF mass spectrometer coupled to an Easy-nLC 1000 II system (Thermo Fisher Scientific). Peptides were loaded on a pre-column (75 µm ID, 6 cm long, packed with ODS-AQ 120 Å–10 µm beads from YMC Co., Ltd) and further packed on an analytical column (75 µm ID, 14 cm long, packed with C18 1.9 µm 100 Å resin from Welch Materials) with a linear reverse-phase gradient from 100% buffer A (0.1% formic acid in H$_2$O) to 28% buffer B (0.1% formic acid in acetonitrile) in 56 (or 71) min at a flow rate of 250 nL/min. The top-12 most intense precursor ions from each full scan (resolution 60,000) were isolated for HCD MS2 (resolution 15,000; normalized collision energy 27) with a dynamic exclusion time of 40 s. Precursors with $3 +$ to $6 +$ charge states were included. Each sample was analyzed a second time using the same parameters, except that $2 +$ precursors were also included for HCD MS2 to find more disulfide bonds in loop-linked peptides.

**Preparation of the SCF(FBXL3)-BS3 and Ca$_v$1.1-SS datasets**. The RAW files of SCF(FBXL3)-BS3 dataset[14] and Ca$_v$1.1-SS dataset[48] were kindly provided by the authors of ref. [14] and ref. [48], respectively. Two complexes were expressed and purified as described therein.

**Preparation of four cell lysate datasets**. The RAW files of E.coli-SS and Human-SS datasets cross-linked by disulfide bonds[13], and E.coli-Leiker and C.elegans-Leiker datasets cross-linked by Leiker[22] were kindly provided by the authors of ref. [13] and ref. [22], respectively. They were prepared as described therein.

**Fragment index construction**. As shown in Fig. 1b, a fragment index is an inverted index data structure, which stores a mapping that associates index keys (mass values) with index values (in silico-digested peptides or their locations in a sequence database file). Supplementary Fig. 17a depicts the workflow of constructing a fragment index for all modified peptides. First, each protein in the specified database was in silico enzymatically digested into peptides, then modified peptides were generated according to user-defined modifications. For each modified peptide, it was represented by its length ($l$), start position ($p$) in the sequence concatenating all proteins in the database, and the modification identifier ($m$)

among all modified peptides from the peptide sequence. Next, for each modified peptide, all neutral masses of $b$, $y$ fragment ions were generated. The ion type information ($t$) and the modified peptide information make up a tetrad code, which was encoded into a 64 bit integer (Supplementary Fig. 17b). Finally, the fragment index was constructed by hashing all integerized masses (original masses × 1000) of fragments and pointing to their parent modified peptides.

Supplementary Fig. 17b depicts an example of encoding and decoding a tetrad code ($t, l, p, m$), which used 1 bit, 7 bits, 46 bits, and 10 bits, respectively. The 1-bit ion type $t$ was able to represent $b$ ion ($t = 1$) or $y$ ion ($t = 0$). The 7-bit length $l$ enabled it to encode peptides with a maximum length of 128 amino acids. The 46-bit start position $p$ gave it possibility to handle database files with a maximum size of 64 TB, which was large enough even for human proteome-scale identification. The 10-bit modification identifier $m$ allowed each peptide sequence to have at most 1024 modified forms. Once the fragment index was constructed, all modified peptides containing one specific fragment mass could be efficiently retrieved in protein databases.

**Fragment index based α-peptide retrieval**. Once the fragment index was constructed, α-peptides were retrieved in four steps as shown in Fig. 1b.

1. Generating query keys. The experimental peaks in MS2 spectra were converted into possible fragment masses as the query keys to query the fragment index.

In the fragment index described above, each modified peptide was indexed by its $b$, $y$ fragment masses. To use this index, we first converted each peak in an MS2 spectrum to four possible fragment masses, assuming that it could be a regular $b$ ion, a regular $y$ ion, an xlink $b$ ion, or an xlink $y$ ion. Xlink $b$ and $y$ ions referred to those that contained a covalently linked β-peptide through the linker, and regular $b$ and $y$ ions were those that do not. The fragment masses were used as the keyword in the next step to retrieve the encoded α-peptide information from the fragment index.

Let $m$ be the mass of a singly charged peak, $m_{H^+}$ be the mass of a proton, $m_{H_2O}$ be the mass of a water molecule, and $M$ be the precursor mass (converted to the singly charged state).

If a peak was a regular $b$ ion, its fragment mass was:

$$m_b = m - m_{H^+}. \quad (1)$$

If a peak was a regular $y$ ion, its fragment mass was:

$$m_y = m - m_{H^+} - m_{H_2O}. \quad (2)$$

If a peak was an xlink $b$ ion, then its complementary ion was a regular $y$ ion and its fragment mass was:

$$m_{\bar{b}} = M - m_{H_2O} - m_b = M - m_{H_2O} - m + m_{H^+}. \quad (3)$$

If a peak was an xlink $y$ ion, then its complementary ion was a regular $b$ ion and its fragment mass was:

$$m_{\bar{y}} = M - m_{H_2O} - m_y = M - m + m_{H^+}. \quad (4)$$

2. Retrieving encoded α-peptides. By querying the fragment index with fragment masses ($\pm 20$ p.p.m. for high-resolution MS2 data), we knew the number of matched peaks, or matched fragment ion counts (MICs), for each retrieved encoded candidate α-peptide. Peptides with MIC = 1 were removed.

3. Decoding α-peptides. From the code of each candidate α-peptide, the modification identifier, the start position, and the length of each retrieved sequence were extracted. As such, the sequences of all α-peptide candidates with MIC ≥ 2 were obtained and the ones whose masses exceeded the precursor mass were removed.

4. Coarse-scoring on α-peptides. For each spectrum, coarse-scoring was performed on α-peptide candidates as described previously (i.e., pre-scoring in pLink 1[10]), and a dynamic list of top-5 results were kept. Coarse-scoring continued if the upcoming candidate sequence had a MIC no smaller than the least MIC of the five candidates in the dynamic list, and the list was updated if the coarse-score of the new candidate was larger than the least coarse-score of the list. For each α-peptide, all possible modifications and all possible cross-linking sites were considered, and only the top scored form was kept.

**Peptide index based β-peptide retrieval**. As shown in Fig. 1c, β-peptide candidates were retrieved from a peptide index, which was a special case of the fragment index. For β-peptides, due to the possible scarcity of matched ions, a peptide index was constructed in the same way as the fragment index, except that the index keys were the masses of intact modified peptides rather than the masses of peptide fragments.

Once the peptide index was constructed, β-peptides were retrieved in three steps as shown in Fig. 1c.

1. Generating query keys. The difference between the precursor mass and the sum of the α-peptide candidate mass and the cross-linker mass was taken as the mass of the β-peptide candidate, which was called as the open mass for α-peptide. Five open masses of five α-peptide candidates were used as the query keywords.

2. Retrieving β-peptides. Each keyword ($\pm 20$ p.p.m. for high-resolution MS2 data) was used to query the peptide index to obtain a list of β-peptide candidates.

3. Fine-scoring on peptide pairs. All retrieved β-peptide candidates were paired with the corresponding α-peptide candidates for fine-scoring as described previously (i.e., fine-scoring in pLink 1[10]). Only the top scored result was kept.

**Re-ranking of PSMs.** pLink 2 inherited the fine-scoring method of pLink 1[10], which was adapted from KSDP[10,61]. However, the score values of different spectra could not be compared without normalization. In pLink 1, normalization was realized through *E*-values. However, for each spectrum, *E*-value calculation required additional 5000 fine-scorings against peptide pairs of random sequences, which was time-consuming. pLink 2 adopted a semi-supervised learning framework to fulfill the normalization task, which was more efficient and flexible.

A machine-learning classifier provides a way to fuse a variety of features and normalize them into a single score. The supervised learning method requires an annotated training set, which is impossible to build, to accommodate different proteomics experiments. Percolator introduced a semi-supervised learning method for peptide identification[43], which eliminated the need to construct a manually curated training set for each experiment. We adopted the same approach in pLink 2 for the identification of cross-linked peptides as shown in Fig. 1d. This contained five steps.

1. Extracting features. There were nine features for cross-linked PSMs: KSDPScore, AlphaIntRatio, BetaIntRatio, AlphaTagRatio, BetaTagRatio, ShortLen, ScoreDiff, ModRatio, and PrecursorErrFreq (Supplementary Table 7). Loop-linked, mono-linked, and regular PSMs contained all above features, except those related to β-peptides. These features were extracted from PSMs and later were combined into one score called SVM score.

2. Selecting training samples. pLink 2 selected positive and negative training samples in an iterative process. In each iteration, the FDR was controlled as

$$\text{FDR} = \frac{N_{TD} - N_{DD}}{N_{TT}}, \tag{5}$$

where $N_{TT}$ denotes the number of PSMs that both α- and β-peptide sequences are from the target database, $N_{DD}$ denotes the number of PSMs that both α- and β-peptide sequences are from the decoy database, and $N_{TD}$ denotes the number of PSMs that one single peptide sequence of the pair is from the target database and the other is from the decoy database. FDRs for intra-protein PSMs and inter-protein PSMs were calculated separately, and if a PSM matched to target and decoy versions of the same protein, this PSM was interpreted as a TD PSM of intra-protein.

FDR threshold was maintained at ≤ 1% while the cutoff value of the KSDP score (for the first iteration) or the SVM score was varied. Positive training samples were PSMs above the cutoff, with both α- and β-peptide sequences from the target database. Negative training samples were formed by the PSMs with either or both α- and β-peptide sequences coming from the decoy database.

3. Updating feature values. The features were separated into static features and dynamic features as shown in Supplementary Table 7. At each iteration, the values of static features were kept unchanged, whereas values of dynamic features were updated based on the positive samples.

4. Training models. An SVM classifier[62] was used to train the model. The maximum number of iterations was set to 5. At any iteration, if the selected positive training set was the same as the one before, the training stopped.

5. Re-scoring and re-ranking. At the end of each iteration, the trained model was used to re-score all top-1 results. We chose L2-loss and L2-regularized linear SVM for the model and the final score was converted to a probability as in logistic regression; the tolerance of termination criterion was set to 0.0001. In addition, L2-regularizer was adopted to avoid overfitting.

The SVM score was given by liblinear with –b option, which means to output probability estimates. The probability was computed using logistic regression and had a range from 0 to 1, which could be interpreted as the probability of a PSM being a random match. The results with scores no more than 0.5 were classified to be positive samples, otherwise negatives.

**Reporting summary.** Further information on experimental design is available in the Nature Research Reporting Summary linked to this article.

## Data availability
A total of 12 datasets were used in the present paper, which were listed in Supplementary Table 2. Eight of 12 datasets were obtained from previous published studies and were referenced appropriately in Supplementary Table 2, and 4 of 12 datasets were prepared in the present paper. Among four new datasets, two were simulated datasets, which were published along with the source code of the simulation method at GitHub [https://github.com/pFindStudio/pLink2/tree/master/pSimXL], and two were $^{15}$N metabolically labeled datasets, which were deposited into ProteomeXchange with Project identifier PXD012109.

## Code availability
pLink 2 was developed in the C/C++ language. The standalone software package can be downloaded at http://pfind.ict.ac.cn/software/pLink/index.html. The source code of the simulation method used to generate simulated spectra, termed as pSimXL, is publicly available at GitHub [https://github.com/pFindStudio/pLink2/tree/master/pSimXL]. Anyone can review and download the source code of pSimXL under the open source GNU General Public License v3.0.

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

## Acknowledgements

We thank Drs. Michael R Hoopmann and Robert L. Moritz for access to CRY2-FBXL3-SKP1 RAW data used for software comparisons; John Hugh Snyder for help with manuscript revision; Shan Lu of the Donglab, and Kun Zhang and Hao Yang of the pFind team for discussion and support. This work was supported by the National Natural Science Foundation of China (21475141 to S.-M.H., 31700727 to C.L., 31470805 to H.C.), the National Key Research and Development Program of China (No. 2016YFA0501300 to S.-M.H.), CAS Interdisciplinary Innovation Team (Y604061000 to S.-M.H.).

## Author contributions

Z.-L.C. and J.-M.M. developed the kernel algorithms of pLink 2. Z.-L.C. developed the interface, analyzed data, and wrote the manuscript. Y.C. prepared the E.coli-SS-$^{15}$N dataset and helped data analysis. J.-L.Y. developed the re-ranking algorithm. R.-Q.F. and S.-B.F. helped analyze disulfide bond datasets. C.L. developed the $^{15}$N quantification tool pQuant. W.-F.Z. helped derive the relationship between global and separate TDA–FDR controls. Y.-H.D. and D.T. prepared the E.coli-Leiker-$^{15}$N dataset. L.W. developed the preprocessing tool pParse. W.-J.Z. and H.C. helped deduce the NaN-FDR on $^{15}$N metabolically labeled datasets. H.C. helped design fragment indexing. R.-X.S. helped revise the manuscript. M.-Q.D. and S.-M.H. directed the study and manuscript writing.

## Additional information

**Competing interests:** The authors declare no competing interests.

