## [Peer Review File · Nature Communications]

Reviewers' comments:

Reviewer #1 (Remarks to the Author):

The manuscript from Chen et al presented nicely a search engine pLink2 for efficient and high confident cross-link identification using non-cleavable cross-linkers. The authors compared pLink2 with several existing and well-recognized cross-link search engines, such as Kojak, StavroX, xQuest and Xi, and concluded that pLink2 outperforms all other software in both sensitivity and accuracy. I highly appreciate this piece of work because I think a systematic evaluation of different cross-linking MS data analysis software is highly beneficial to the cross-linking community. While I find the overall result is convincing, I would like to ask the authors to further address my following concerns, which I think is necessary for the thorough evaluation of a newly developed software tool.

Major points:

1. The authors chose 9 different software to compare to pLink2 using a simulated dataset and further chose 2 out of 9 best performing software (Kojak and pLink1) for further evaluation using a synthetic dataset. Although the authors described in the supplementary data how the simulated dataset was generated, I find it is difficult to fully exclude the possibility that this simulated dataset is in favor of the software that the authors have developed (pLink1 and pLink2). Therefore, I would like the authors to include more software (for instance StavroX and Xi, which are widely used in different biological applications) for a comparison with pLink2 using the synthetic dataset.
2. I would like the authors to also include the XlinkX node in Proteome Discoverer for comparison using both simulated and synthetic datasets. Although this search engine is mainly focused on cleavable cross-linkers, it also supports data analysis of non-cleavable cross-linkers.
3. In the result section "Credibility evaluation using 15 277 N metabolically labeled datasets", I am not sure if I understand correctly how the experiment was designed (page 9, lines 278-286). As far as I understand, the authors first perform cross-link identification for any given peptide features that can be detected in MS1 and then looks for the existence of MS1 signals at the m/z of the corresponding N15 labeled peptides. If this is the case, this approach will only evaluate the performance of MS1 feature detection (i.e., if a cross-link is detected from a real peptide signal) but does not necessarily make assessment whether a cross-link identification is correct or not. I would like the authors to give further explanations on experiment design and data evaluation in this section.

Minor points:

1. I would like the authors to change the question sentences on Page 3 lines 80-83 to statement sentences.
2. Please rephrase the following sentence on page 8 lines 256-257.
3. Page 12, lines 383-385, "Despite this, consider that many studies have shown that the FDR of intersection results drops rapidly". Please rephrase this sentence.

Reviewer #2 (Remarks to the Author):

In their paper entitled "pLink 2: A high-speed search engine with systematic evaluation for proteome-scale identification of cross-linked peptides", Chen et al. propose a new software tool for the confident identification of cross-linked peptides in complex mixtures analyzed by high resolution mass

spectrometry. They benchmark their new tool, in particular against pLink1 and Kojak using different types of datasets and show that pLink2 overperforms in speed, precision and sensitivity.

Even if the paper looks interesting, there are major issues that precludes its publication in Nature Communications. The first and major one relies in the fact that this is a very technical paper, with a lot of details on the informatics behind the software and Nature Communications does not appear as the best journal to publish such results. A more bioinformatics-centric journal such as Bioinformatics would be a better place for such paper.

A second issue relies on the fact that the datasets that were used are not described, do not represent a real sample, and thus there maybe a bias in favor of pLink2 using such datasets.

Finally, the first part of the software is similar to another one, which is not mentioned here which is MassSpecStudio 2.0 that works also well for entire proteomes. This is clearly not the case for all software tools dedicated to proteome-wide analysis of cross-linked data and thus a comparison with this particular tool would be very useful.

Response to Reviewers' comments

We thank the reviewers for their insightful and very constructive comments, and have revised our manuscript accordingly. We are also grateful to the editor for giving us a chance to submit a revision. Our replies to specific comments and suggestions are as follows.

Reviewers' comments:

Reviewer #1 (Remarks to the Author):

The manuscript from Chen et al presented nicely a search engine pLink2 for efficient and high confident cross-link identification using non-cleavable cross-linkers. The authors compared pLink2 with several existing and well-recognized cross-link search engines, such as Kojak, StavroX, xQuest and Xi, and concluded that pLink2 outperforms all other software in both sensitivity and accuracy. I highly appreciate this piece of work because I think a systematic evaluation of different cross-linking MS data analysis software is highly beneficial to the cross-linking community. While I find the overall result is convincing, I would like to ask the authors to further address my following concerns, which I think is necessary for the thorough evaluation of a newly developed software tool.

Reply 1-1:

Thank you very much for your positive comments.

During the development of pLink 2, while we were happy to see the significant improvement of search speed, we were concerned with whether or not the identified PSMs were credible enough. Luckily, many evaluation methods had been proposed in related fields, such as the use of simulated datasets, synthetic datasets, ¹⁵N metabolically labeled datasets, and entrapment databases. Therefore, we adapted these methods to systematically evaluate the precision and sensitivity of pLink 2 and compare among different search engines.

While we were preparing the pLink 2 manuscript, we pondered whether we should emphasize the fragment index or the systematic evaluation method. Although they were both important for pLink 2, we finally decided to emphasize the systematic evaluation in the main text and put the fragment index in the supplementary information. Because only when the search engine is credible can we discuss speed. Additionally, through this manuscript, we want to raise concern about systematic evaluation of software tools, which is especially lacking in the field of cross-linking mass spectrometry (CXMS).

Major points:

1. The authors chose 9 different software to compare to pLink2 using a simulated dataset and further chose 2 out of 9 best performing software (Kojak and pLink1) for further evaluation using a synthetic dataset. Although the authors described in the supplementary data how the simulated dataset was generated, I find it is difficult to

fully exclude the possibility that this simulated dataset is in favor of the software that the authors have developed (pLink1 and pLink2). Therefore, I would like the authors to include more software (for instance StavroX and Xi, which are widely used in different biological applications) for a comparison with pLink2 using the synthetic dataset.

Reply 1-2:

We are sorry that the method used to generate simulated spectra was not described clearly and had raised concern about the fairness of the simulated datasets. In the revision, we modified the method to generate simpler simulated spectra, made the source code public, and compared all ten search engines using the two synthetic datasets. These three revisions are elaborated as follows.

Firstly, we added more detailed information about the method used to generate simulated spectra (Supplementary Note 8 and copied below). In summary, the method is simple and contains five steps:

1. *In silico* digest proteins and generate modified peptides;
2. Randomly select two modified peptides as the α - and β -peptide;
3. Randomly assign a charge and then determine the m/z of the precursor;
4. Generate a rich set of fragment ion peaks;
5. Merge adjacent fragment ion peaks.

In the original manuscript, fragment ions b^{1+} , b^{2+} , y^{1+} , y^{2+} , y^{3+} , a^{1+} , and a^{2+} were used. In the revised manuscript, in order to generate simpler simulated spectra, the fragment ions y^{3+} , a^{1+} , and a^{2+} were excluded, for they are hardly used by other search engines; only the fragment ions b^{1+} , b^{2+} , y^{1+} , and y^{2+} were included, which are used by all search engines. In order to generate higher-quality simulated spectra, the occurrence probability of each peak type was increased to be much higher than that in the synthetic datasets (Fig. S16d), resulting in more fragment ion peaks in the simulated spectra than in the spectra of synthetic peptides.

Supplementary Note 8. The method used to generate simulated spectra

The Simulated-BS3 dataset consists of cross-linked, loop-linked, mono-linked, and regular MS2 spectra, 2,500 for each type, resulting in 10,000 MS2 spectra in total. As there are no mono-linked peptides in a disulfide bond sample, the Simulated-SS dataset only consists of 2,500, 2,500, and 5,000 for cross-linked, loop-linked, and regular MS2 spectra, respectively, resulting in 10,000 MS2 spectra in total.

Once the protein database, cross-linker, and modifications are specified (Table S7), cross-linked MS2 spectra are generated in five steps as detailed below.

1. Each protein in the specified database is *in silico* enzymatically digested into peptides, and then modified regular peptides are generated according to user-defined modifications.
2. Two modified regular peptides, containing cross-linkable residues, are randomly selected as the α -peptide and the β -peptide.
3. Randomly assign a precursor charge state of +3, +4, +5, or +6 with probability 55%, 35%, 7%, or 3%, respectively. The charge distribution is obtained according to the synthetic datasets^{1, 16}. Hence, precursor m/z can be

determined by combining the charge with α -peptide mass, β -peptide mass and linker mass.

- a. To better simulate the real situation, Gaussian error $e_1 \sim N(0, 3^2)$ ppm is added to the precursor m/z, and make sure that e_1 does not exceed 9.9 ppm.
4. Theoretical m/z of fragment ions b^{1+} , b^{2+} , y^{1+} , y^{2+} are calculated. The intensity of each ion type is set according to the average intensity of this ion type on the synthetic datasets (Average intensity in the Fig. S16d). These theoretical ions with full m/z and intensity information make up the main fragment ion peaks. To better simulate the real situation, four types of noises are added to each fragment ion peak:
 - a. A fragment ion peak is added with occurrence probability $\text{Prob}_{\text{occur}}$ in Fig. S16d.
 - b. Gaussian error $e_2 \sim N(0, 3^2)$ ppm is added to the m/z of the peak.
 - c. Add one noise peak with probability 20%. The m/z of the noise peak equals to the m/z of the fragment ion peak adding Gaussian error $e_3 \sim N(0, 6.5^2)$ ppm, the intensity of the noise peak equals to 10% of the intensity of the fragment ion peak.
 - d. The first and second isotopic peaks of the fragment ion peak are added to the spectrum to help search engines determine the charge state of the fragment ion peak. The intensities of isotopic peaks are calculated using the EMASS algorithm¹⁷.
5. Finally, merge adjacent peaks with approximately the same m/z ($\Delta\text{m/z} < 1\text{E-5 Th}$).

In Step 4a, we calculate the occurrence probabilities ($\text{Prob}_{\text{occur}}$) of ion types b^{1+} , b^{2+} , y^{1+} , and y^{2+} for the Synthetic-BS3 dataset (Fig. S16a) and the Synthetic-SS dataset (Fig. S16b). The $\text{Prob}_{\text{occur}}$ is the number of matched peaks of an ion type divided by the total number of theoretical peaks of that ion type, which means the occurrence probability of that ion type. On average, y^{1+} has the highest occurrence probability (0.60), while the occurrence probabilities of other three ion types lie between 0.25 and 0.33 (Fig. S16c). In order to generate higher-quality simulated spectra, we increase the occurrence probability of y^{1+} to 0.8, and increase the occurrence probabilities of other three ion types to 0.5 (Fig. S16d), making the simulated spectra contain more fragment ion peaks than the spectra of synthetic peptides.

Fig. S17 shows two examples of simulated BS3 and SS cross-linked spectra. Loop-linked, mono-linked, and regular spectra are generated in the similar way except that only one modified peptide needed to be randomly selected from all modified peptides in step 2.

As both α - and β - peptides are theoretically fragmented better than synthetic peptides, not containing any complicated ions such as internal ions, the simulated spectra are rather simpler compared with the real-world spectra. In our study, we firstly use simulated datasets to evaluate ten established cross-linked peptide search engines, and only those passing this qualification test will proceed to participate in the following comparisons.

The source code of the simulation method used to generate simulated spectra, termed as pSimXL, is publicly available at GitHub: <https://github.com/pFindStudio/pLink2/tree/master/pSimXL>. Anyone can review and download the source code of pSimXL under the open source GNU

General Public License v3.0. More importantly, any search engine for identification of cross-linked peptides can use pSimXL to debug and improve performance. We believe that pSimXL will be highly beneficial to the CXMS community.

Table S7. The parameters used for generating and searching two simulated datasets

Items	Settings
Database	First 100 proteins in E. coli database
Spectra	10,000 spectra (2,500 cross-linked spectra and 7,500 non -cross-linked spectra)
Cross-linker	BS3 for Simulated-BS3 and SS for Simulated-SS
Enzyme	Trypsin
Max Missed Cleavage Sites	2
Peptide Mass Range	[600, 6,000] Da
Peptide Length Range	[6,60]
Precursor Tolerance	± 10 ppm
Fragment Tolerance	± 20 ppm
Fixed Modifications	Carbamidomethylation (C) for Simulated-BS3
Variable Modifications	Oxidation (M) for Simulated-BS3 and Nethylmaleimide(C) for Simulated-SS
Max Modified Sites	3

a Synthetic-BS3			b Synthetic-SS		
Ion type	Prob _{occur}	Average intensity	Ion type	Prob _{occur}	Average intensity
b^{1+}	0.19	0.12	b^{1+}	0.30	0.10
b^{2+}	0.23	0.08	b^{2+}	0.24	0.06
y^{1+}	0.71	0.22	y^{1+}	0.48	0.20
y^{2+}	0.37	0.14	y^{2+}	0.29	0.09

c Average of Synthetic-BS3 and Synthetic-SS			d Simulated-BS3 and Simulated-SS		
Ion type	Prob _{occur}	Average intensity	Ion type	Prob _{occur}	Average intensity
b^{1+}	0.25	0.11	b^{1+}	0.50	0.11
b^{2+}	0.24	0.07	b^{2+}	0.50	0.07
y^{1+}	0.60	0.21	y^{1+}	0.80	0.21
y^{2+}	0.33	0.12	y^{2+}	0.50	0.12

Fig. S16. The fragmentation characteristics of synthetic datasets and simulated datasets. **a)** The occurrence probabilities and average intensities of ion types b^{1+} , b^{2+} , y^{1+} , and y^{2+} for the Synthetic-BS3 dataset. The occurrence probability is the number of matched peaks of an ion type divided by the total number of theoretical peaks of that ion type. The average intensity is the average of normalized intensity of matched peaks belonging to the same ion type (normalized to the base peak intensity). **b)** Similar to a), but for the Synthetic-SS dataset. **c)** The average of a) and b). **d)** The occurrence probabilities and average intensities used to generate simulated spectra. Compared with synthetic datasets, the occurrence probability of y^{1+} is increased to 0.8 and the occurrence probabilities of other three ion types are increased to 0.5, making the simulated spectra contain more fragment ion peaks than the spectra of synthetic peptides.

With the revised simulation method, we randomly generated three different Simulated-BS3 datasets and three different Simulated-SS datasets, and evaluated all ten search engines again. For each of the ten search engines, Table 1 and Supplementary Table 3 (both were copied below) show the averaged performance on three Simulated-BS3 datasets and on three Simulated-SS datasets, respectively. The results were similar to those in the initial version of this manuscript: pLink 2, pLink 1, and Kojak were still the top-3 highest-sensitivity search engines.

Table 1. The performance of ten search engines on the Simulated-BS3 dataset^a

Search engine	Search strategy	Sensitivity (%)	Precision (%)	Run time (Min)	Selected
xQuest ^b	Exhaustive	-	-	-	×
Xilmass ^c	Exhaustive	-	-	-	×
Xolik	Exhaustive	40.7	93.7	0.7	×
StavroX	Exhaustive	50.4	78.6	363.9	×
Xi	Open	62.5	59.1	9.3	×
MetaMorpheusXL	Open	71.0	97.7	0.3	×
Protein Prospector	Open	78.6	97.2	16.9	×
Kojak	Open	85.3	97.8	1.7	✓
pLink 1	Open	99.8	99.8	12.3	✓
pLink 2	Open	99.9	100.0	0.5	✓

a For sensitivity, precision, and run time, the average values obtained using three randomly generated Simulated-BS3 datasets were shown.

b xQuest threw an exception “Illegal division by zero at

/home/xqxp/xquest/V2_1_1/xquest/bin/compare_peaks3.pl line 2246” and did not report any results.

c Xilmass threw an exception “java.lang.OutOfMemoryError: GC overhead limit exceeded” and did not report any results.

Supplementary Table 3. The performance of ten search engines on the Simulated-SS dataset^a

Search engine	Search strategy	Sensitivity (%)	Precision (%)	Run time (Min)	Selected
xQuest ^b	Exhaustive	-	-	-	×
Xilmass ^c	Exhaustive	-	-	-	×
Xolik	Exhaustive	19.9	97.3	0.4	×
MetaMorpheusXL	Open	57.8	98.1	0.2	×
StavroX	Exhaustive	61.1	47.5	42.6	×
Protein Prospector	Open	64.5	97.9	18.8	×
Xi	Open	68.6	95.5	0.9	×
Kojak	Open	79.7	96.7	1.2	✓
pLink 1	Open	99.8	92.6	8.9	✓
pLink 2	Open	99.9	100.0	0.7	✓

a For sensitivity, precision, and run time, the average values obtained using three randomly generated Simulated-SS datasets were shown.

b xQuest did not report any results.

c Xilmass did not support the disulfide bond cross-linker and could not set user defined cross-linkers.

Secondly, we published the source code of our simulation method, termed as pSimXL, which is publicly available at

GitHub: <https://github.com/pFindStudio/pLink2/tree/master/pSimXL>. Anyone can

review and download the source code of pSimXL under the open source GNU

General Public License v3.0. More importantly, any search engine for identification of cross-linked peptides can use pSimXL to debug and improve performance. We believe that pSimXL will be highly beneficial to the CXMS community.

Finally, following reviewer’s advice, we compared pLink 2 with all other nine search

engines using two Synthetic datasets (Supplementary Note 1 and copied below). The conclusions are still the same: pLink 2, pLink 1, and Kojak were the top-3 search engines for proteome-scale cross-link identification.

Supplementary Note 1. Evaluate the performance of ten search engines using synthetic datasets

In the main text, according to the evaluation result on the simulated datasets, only the top-3 highest-sensitivity search engines, namely pLink 2, pLink 1, and Kojak, were selected for further evaluation using synthetic datasets. In this supplementary note, we tested all ten search engines on the synthetic datasets.

Take the Synthetic-BS3 dataset as an example. First of all, ten search engines searched all 2,077 spectra against the original database. As the sensitivities of ten search engines varied greatly from one another, the intersection of identifications from all ten search engines would be very small (Original database in Fig. S1). We thus only took the intersection of identifications from Kojak, pLink 1, and pLink 2 as a new and fair standard dataset for searching against increasingly larger databases generated by appending the *E. coli*, worm, or human database as an entrapment to the original database. The new and fair Synthetic-BS3 dataset consists of 904 PSMs (same as in Fig. 2a).

Fig. S1 shows that only six of ten search engines successfully finished searching against the *E. coli* entrapment database, they were Xolik, MetaMorpheusXL, Kojak, PP, pLink 1, and pLink 2 sorted by sensitivity in ascending order. Furthermore, the sensitivities of search engines decreased as the database size increased (Fig. S2a), while the precisions were relatively stable (Fig. S2b). Evaluations using three different entrapment databases showed that pLink 2 achieved the highest sensitivity, followed closely by pLink 1. In contrast, the sensitivities of PP, Kojak, MetaMorpheusXL, and Xolik decreased significantly, especially for MetaMorpheusXL and Xolik, whose sensitivities decreased down to less than 10% with both the worm and the human entrapment databases. The results obtained for the Synthetic-SS dataset were similar to those obtained for the Synthetic-BS3 dataset, except that MetaMorpheusXL and Xolik both threw an “OutOfMemoryError” exception and failed to finish searching against the worm and the human entrapment databases (Figs. S3 and S4).

Search engines were also compared in terms of computing time. Where possible, 8 threads were used for each search engine. The normalized computing times for ten search engines (Windows Server, Intel Xeon E5-2670 CPU with 32 cores, 2.6 GHz, 128 GB RAM) are shown in Table S1. Most search engines did not finish searching against the worm and the human entrapment databases. pLink 2 was the fastest one that passed all evaluations and achieved the highest sensitivities. Although Xolik was faster than pLink 2, it had very low sensitivity (Fig. S1) and failed to control the memory usage when searching the disulfide bond data against big databases (Fig. S3).

Generally speaking, xQuest, Xilmass, StavroX, and Xi do not support proteome-scale cross-linked peptide identification, while the other six do. Search engines in the latter group can be further categorized into three types according to sensitivity: Xolik and MetaMorpheusXL have extremely low sensitivities; PP and Kojak have moderately higher sensitivities; pLink 1 and pLink 2 have the highest sensitivities. Performance comparison among all ten search engines using synthetic datasets shows that pLink 2, pLink 1, and Kojak were indeed the top-3

highest-sensitivity search engines, which confirmed the conclusion drawn from the evaluation using simulated datasets.

Fig. S1. On the Synthetic-BS3 dataset, the numbers of correctly identified PSMs by each search engine, when searching 904 spectra against the original database plus the *E. coli*, worm, and human entrapment databases.

Fig. S2. On the Synthetic-BS3 dataset, the **a**) sensitivity and **b**) precision of ten

search engines when searching against original database plus entrapment databases of different sizes. Search engines were sorted by sensitivity in original database in descending order.

Fig. S3. On the Synthetic-SS dataset, the numbers of correctly identified PSMs by each search engine, when searching 1,911 spectra against the original database plus the *E. coli*, worm, and human entrapment databases.

Fig. S4. On the Synthetic-SS dataset, the **a)** sensitivity and **b)** precision of ten search engines when searching against original database plus entrapment databases of different sizes. Search engines were sorted by sensitivity in original database in descending order.

Table S1. The normalized computing time of ten search engines on two synthetic datasets

Dataset		xQuest	Xilmass	Xolik	MetaMorpheusXL	Xi	PP	Kojak	StavroX	pLink 1	pLink 2		
Synthetic-BS3	Original		5.5	1.8	2.0	56.8	26.0	2.3	1.8	35.5	1.0	*(0.1)	
	+ E. coli	b	c	0.5	5.1	c	7.7	1.7	f	45.4	1.0	(0.5)	
	+ Worm			0.3	35.2		11.6	2.4		50.8	1.0	(3.5)	
	+ Human			0.2	56.7		e	2.1		38.3	1.0	(6.4)	
Original	d			c	c		c	2.0		1.2	4.7	35.5	2.6
+ E. coli		0.6	5.9			17.0		3.3	60.2	1.0	(1.2)		
+ Worm		e	e			10.4		3.8	64.3	1.0	(17.7)		
+ Human						69.7		1.0	(31.9)				
Average ^a	-	-	0.9	17.7	-	18.0	2.8	-	-	49.6	1.0	-	

a The average of normalized computing times was calculated only for search engines that support proteome-scale cross-linked peptide identification.

b xQuest threw an exception "Illegal division by zero at /home/xqxp/xquest/V2_1_1/xquest/bin/compare_peaks3.pl line 2246" and did not report any results.

c These search engines threw an exception "OutOfMemoryError" and did not report any results.

d Xilmass did not support the disulfide bond cross-linker and could not set user defined cross-linkers.

e PP threw a "File open failure" exception and failed to start searching.

f StavroX did not finish searching within one week.

g The real search times in minutes are shown in parentheses.

2. I would like the authors to also include the XlinkX node in Proteome Discoverer for comparison using both simulated and synthetic datasets. Although this search engine is mainly focused on cleavable cross-linkers, it also supports data analysis of non-cleavable cross-linkers.

Reply 1-3:

Thank you for your advice. We have included the XlinkX node in Proteome Discoverer 2.3 for comparison, which was placed in Supplementary Note 4 and copied below. The main conclusions were that pLink 2 identified four times more cross-linked PSMs than XlinkX, and those uniquely identified by pLink 2 had lower percentage of abnormal quantitation ratios (NaN-ratios) and hence were more credible. Furthermore, pLink 2 was about 30 times faster than XlinkX, and XlinkX cannot search against databases with more than 1,500 proteins.

Supplementary Note 4. Evaluate the performance of XlinkX

The software XlinkX¹⁰⁻¹¹ is a search engine designed mainly for analysing data from MS-cleavable cross-linkers, but it also supports data analysis of non-cleavable cross-linkers. We have tried the latest XlinkX node in Proteome Discoverer 2.3 using the xlinkx23_noncleavable.pdAnalysis template file provided on the Heck lab's website (<https://www.hecklab.com/software/xlinkx/>). We found that XlinkX can analyse RAW files but cannot analyse MGF files, and therefore we cannot evaluate XlinkX using the simulated and synthetic datasets. We finally evaluated the performance of XlinkX using ¹⁵N metabolically labeled datasets and entrapment databases, for which RAW files are available.

Credibility evaluation using ¹⁵N metabolically labeled datasets

For the *E.coli-Leiker-¹⁵N* dataset. As XlinkX only supports up to 1,500 proteins when analysing non-cleavable cross-linker data

(<https://www.hecklab.com/software/xlinkx/>), the database for XlinkX search contained only 1,284 proteins identified from RAW files of the sample using a regular peptide search engine, pFind¹² (restricted search mode with parameters in Supplementary Table 5 except that the FDR was set as 5% at peptide level). The parameters for XlinkX search were the same as those for pLink 2 (Supplementary Table 5). pLink 2 also searched against the same restricted database of 1,284 proteins, so that results could be fairly compared.

XlinkX identified a total of 1,255 cross-linked PSMs with 1.4% NaN-ratios, while pLink 2 identified fourfold of that (5,450) with only 0.6% NaN-ratios. Furthermore, PSMs uniquely identified by pLink 2 had much lower percentage of NaN-ratios compared with PSMs uniquely identified by XlinkX (Figs. S7a and S7b). Results at peptide pair level were similar to those at PSM level (Figs. S7c and S7d), showing the superiority of pLink 2 both in sensitivity and precision. Finally, for cross-linked results obtained by pLink 2 in restricted database (Fig. S7) and in *E. coli* whole proteome database (Fig. 3), the proportion of intersection set to union set is 91% and 87% respectively for PSMs and peptide pairs, showing the high consistency of pLink 2 when searching against the restricted database and the *E. coli* whole proteome database.

Fig. S7. Compare pLink 2 with XlinkX on the *E.coli*-Leiker-¹⁵N dataset. At PSM level, the Venn diagram for **a**) inter-protein and **b**) intra-protein PSMs identified by XlinkX and pLink 2. Numbers in parentheses indicate the number and the percentage of NaN-ratio results that lie in the corresponding region. For example, 56 (15, 26.8%) means that XlinkX uniquely identified 56 inter-protein PSMs, of which 15 were NaN-ratios, accounting for 26.8% of 56. **c**) and **d**) are similar to a) and b) respectively, but at peptide pair level.

For the E.coli-SS-¹⁵N dataset. Similarly, the database for XlinkX and pLink 2 searches contained only 1,181 proteins identified from RAW files of the sample using

pFind. XlinkX identified a total of 6,19 cross-linked PSMs with 0% NaN-ratios, while pLink 2 identified sevenfold of that (4,448) with 1.0% NaN-ratios. Although XlinkX did not identify any NaN-ratio PSM, it identified many fewer cross-linked PSMs than pLink 2, and 96% of them were covered by pLink 2 (Fig. S8), indicating the low sensitivity of XlinkX when analysing non-cleavable cross-linker data at a proteome scale of ~1,200 proteins.

Fig. S8. Compare pLink 2 with XlinkX on the E.coli-SS-¹⁵N dataset. At PSM level, the Venn diagram for **a**) inter-protein and **b**) intra-protein PSMs identified by XlinkX and pLink 2. **c**) and **d**) are similar to a) and b) respectively, but at peptide pair level.

Credibility evaluation using entrapment databases

The entrapment database method used in the main text took the intersection of cross-linked PSMs identified by Kojak, pLink 1, and pLink 2 as a new and fair standard dataset to search against entrapment databases of different sizes. However, the intersection spectra could not be a RAW file, thus could not be analysed by XlinkX. Nevertheless, XlinkX could search the original RAW files against entrapment databases, and the intersection PSMs identified by Kojak, pLink 1, and pLink 2 could serve as a ground truth.

Take the SCF(FBXL3)-BS3 dataset as an example. XlinkX searched three original RAW files against the original database plus increasing number of proteins from the *E. coli* database. As XlinkX only supports up to 1,500 proteins when analysing non-cleavable cross-linker data, the entrapment database for XlinkX search contained at most 1,000 proteins from the *E. coli* database. Furthermore, XlinkX threw an “OutOfMemoryError” exception when five variable modifications were set as Supplementary Table 6, thus only Oxidation[M] was set as a variable modification. All other parameters were the same as those in Supplementary Table 6. Among the 850 cross-linked PSMs consistently identified by Kojak, pLink 1, and pLink 2 (Fig. 4 in the main text), 846 PSMs were not modified or modified only by Carbamidomethyl[C] or Oxidation[M]. Therefore, we took the 846 PSMs as a new

and fair standard dataset (named intersection-846) to assess sensitivity and precision of XlinkX when searching against entrapment databases of different sizes. An identification was deemed correct if it is identical to an annotated PSM in the intersection-846; otherwise it was considered incorrect.

Fig. S9 shows that although the sensitivities and precisions of pLink 2 and XlinkX kept stable along the entrapment database size increased, the sensitivity of XlinkX was less than 40%, much lower than that of pLink 2 (~99%).

For the Ca_v1.1-SS dataset, all parameters were the same as those in Supplementary Table 6, including two variable modifications. Therefore, we took the 836 PSMs consistently identified by Kojak, pLink 1, and pLink 2 as a new and fair standard dataset (named intersection-836, same as in Supplementary Fig. 12a) to assess sensitivity and precision of XlinkX when searching against entrapment databases of different sizes. Evaluation results were similar to those obtained for the SCF(FBXL3)-BS3 dataset (Fig. S10). A possible reason for the stable sensitivity and precision might be that the size of entrapment database was too small, and if a whole proteome database was added as an entrapment database, more evident changes might be seen (Fig. 4 in the main text).

Fig. S9. Performance evaluation on the SCF(FBXL3)-BS3 dataset. The **a)** sensitivity and **b)** precision of pLink 2 and XlinkX when searching three original RAW files against the original database plus increasing number of proteins from the *E. coli* database. The 846 PSMs consistently identified by Kojak, pLink 1, and pLink 2 were used as a new and fair standard dataset to assess sensitivity and precision.

Fig. S10. Performance evaluation on the Ca_v1.1-SS dataset. The **a)** sensitivity and **b)** precision of pLink 2 and XlinkX when searching two original RAW files against the original database plus increasing number of proteins from the *E. coli* database. The 836 PSMs consistently identified by Kojak, pLink 1, and pLink 2 were used as a new and fair standard dataset to assess sensitivity and precision.

Speed evaluation of XlinkX

XlinkX and pLink 2 were also compared in terms of computing time. Where possible, 8 threads were used for each comparison. The normalized computing times for XlinkX and pLink 2 (Windows 10, Intel Xeon E3-1241 CPU with 8 cores, 3.5 GHz, 16 GB RAM) are shown in Table S2. On average, pLink 2 was 30 times faster than XlinkX when analysing data from non-cleavable cross-linkers. For benchmarking in the entrapment database method, the speedup of pLink 2 increased as the entrapment database size increased, showing the high efficiency of pLink 2, especially at a proteome scale.

Table S2. The normalized computing time of XlinkX and pLink 2 on four datasets

Dataset	XlinkX	pLink 2	
E.coli-Leiker- ¹⁵ N	22.7	1.0	^a (67.4)
E.coli-SS- ¹⁵ N	10.4	1.0	(65.2)
SCF(FBXL3)-BS3	Original	31.4	1.0 (12.2)
	+ E.coli 200	34.6	1.0 (19.2)
	+ E.coli 400	36.2	1.0 (26.2)
	+ E.coli 600	36.9	1.0 (32.2)
	+ E.coli 800	43.9	1.0 (33.9)
	+ E.coli 1000	49.3	1.0 (37.2)
Cav1.1-SS	Original	40.0	1.0 (2.9)
	+ E.coli 200	18.2	1.0 (12.6)
	+ E.coli 400	20.4	1.0 (16.8)
	+ E.coli 600	23.1	1.0 (23.5)
	+ E.coli 800	25.1	1.0 (28.5)
	+ E.coli 1000	27.1	1.0 (35.5)
Average	29.9	1.0	-

^a The real search times in minutes are shown in parentheses.

3. In the result section “Credibility evaluation using 15 277 N metabolically labeled datasets”, I am not sure if I understand correctly how the experiment was designed (page 9, lines 278-286). As far as I understand, the authors first perform cross-link identification for any given peptide features that can be detected in MS1 and then looks for the existence of MS1 signals at the m/z of the corresponding N15 labeled peptides. If this is the case, this approach will only evaluate the performance of MS1 feature detection (i.e., if a cross-link is detected from a real peptide signal) but does not necessarily make assessment whether a cross-link identification is correct or not. I would like the authors to give further explanations on experiment design and data evaluation in this section.

Reply 1-4:

We apologize that the credibility evaluation method using ¹⁵N metabolically labeled datasets was not described clearly. In the revision, we explained it in greater detail and by example.

Firstly, we have revised the experimental design and added more details to explain how to use ¹⁵N metabolically labeled datasets to evaluate the credibility of search engines (page 9, lines 288-301 in the revised manuscript and copied below). Step-by-step description of the evaluation method was added to the caption of Fig. 3a (copied below).

Then, two 1:1 ^{15}N metabolically labeled datasets, E.coli-Leiker- ^{15}N and E.coli-SS- ^{15}N , were prepared and used to evaluate the precision of search engines using the percentage of PSMs with invalid quantitation ratios (Fig. 3a). Given the experimental design, if a spectrum is identified as an unlabeled peptide or peptide pair, the corresponding ^{15}N -labeled precursor ion should be observed in the MS1 scans. The mass distance between the unlabeled and ^{15}N -labeled precursors is determined by the number of nitrogen atoms. Generally speaking, a falsely identified peptide or peptide pair carries an unexpected number of nitrogen atoms, thus the chance is small that the calculated ^{15}N -labeled version happens to have matched signals in the experimental spectra. As such, falsely identified peptides or peptide pairs will most likely have invalid quantitation ratios, *i.e.* Not-a-Number (NaN), when quantified by pQuant⁴⁷ (Supplementary Fig. 5). The percentage of NaN-ratios reflects the collective confidence level of given PSM identifications; a smaller percentage of NaN-ratios corresponds to higher precision.

Fig. 3a. Experimental design of the E.coli-Leiker- ^{15}N dataset. The unlabeled and ^{15}N metabolically labeled *E. coli* lysates were cross-linked separately, mixed at a 1:1 ratio, digested with trypsin, and analysed by LC-MS/MS. The dataset was searched only for the unlabeled peptides using different search engines, and the identification results were passed to pQuant to quantify the intensity ratio of the ^{15}N -labeled precursor to the unlabeled precursor. Lastly, the precision of identifications was investigated by checking the percentage of NaN-ratio PSMs and peptides.

Secondly, we have added one example to demonstrate validation of search results using ^{15}N metabolically labeled datasets (Supplementary Fig. 5 and copied below).

Thirdly, we want to explain more deeply about the experimental design of the evaluation method. Given the MS data generated from the ^{15}N metabolically labeled sample, the whole experiment contains three steps:

1. *MS1 feature detection.* As the precursor m/z of one MS2 given by the instrument is not always the monoisotopic m/z due to the peak interference within the isolation window, some search engines adopted MS1 feature detection methods to calibrate and extract multiple precursor ions for each MS2 scan. For example, pParse (*Yuan et al., Proteomics 2012*) and Hardklor (*Hoopmann et al., Anal. Chem. 2007*) are integrated by pLink 2 and Kojak, respectively. Both pParse and Hardklor do not detect whether a precursor is a cross-link or not, they just perform deconvolution and report the monoisotopic m/z and charge of a precursor, hence the monoisotopic peak could be a cross-linked peptide, a regular peptide, or any other signal.
2. *Identification.* Given one MS2 scan and the calibrated precursor mass, search engines do not know whether it is a cross-linked peptide or not, and hence they search all possibilities including cross-linked, loop-linked, mono-linked, and regular peptides, and only keep the highest fine-scored candidate as the identification for the MS2 scan.
3. *Quantitation.* After identification, we use pQuant (*Liu et al., Anal. Chem. 2014*) to perform a routine ^{15}N -labeled quantitation analysis. Given the unlabeled identification and its m/z, the corresponding ^{15}N -labeled m/z is deduced according to the number of nitrogen atoms in the identification, pQuant extracts the unlabeled and ^{15}N -labeled MS1 signals in MS1 scans and calculates the ^{15}N -labeled/unlabeled quantitation ratio. If the ^{15}N -labeled signals are not observed in MS1 scans, pQuant outputs a NaN ratio.

Therefore, a NaN quantitation ratio could be caused by three factors: incorrect monoisotopic peak detected by MS1 feature detection method (see Fig. R1 below), incorrect identification by search engine (see Supplementary Fig. 5 above), and incorrect quantitation ratio calculated by pQuant. We have shown that the percentage of NaN-ratio PSMs in regular peptide identifications was at most 0.3% (Supplementary Fig. 6), which means at most 0.3% NaN-ratio PSMs were caused by pQuant. However, the percentages of cross-linked PSMs identified by Kojak and pLink 1 that had NaN-ratios were 6.4% and 3.8%, respectively (Fig. 3 in main text), which were much larger than 0.3%. The percentage of cross-linked PSMs identified by pLink 2 that had NaN-ratios was only 0.5%, which was very close to the percentage of regular PSMs having NaN-ratios (at most 0.3%), indicative of good performance of pLink 2 in both MS1 feature detection and identification.

As such, the percentage of NaN-ratios can be used to evaluate the performances of both MS1 feature detection and identification. Since most search engines have integrated their own MS1 feature detection methods, such as pParse in pLink 2 and Hardklor in Kojak, we used the percentage of NaN-ratios to roughly evaluate the overall performances of search engines. Actually, the ^{15}N metabolically labeled datasets coupled with pQuant had been used to evaluate the credibility of glycopeptide search engine pGlyco 2.0 (*Liu et al., Nat. Commun. 2017*) and regular peptide search engine Open-pFind (*Chi et al., Nat. Biotechnol. 2018*), and the method used here is the same.

Finally, we want to apologize that there was one error of description in the “Preparation of the E.coli-Leiker-¹⁵N dataset” subsection of the Datasets section in the initial version of this manuscript. We have mistaken the order of the cross-linking reaction and the mixing process in the description. The actual procedure was that the cross-linking reaction was before the mixing process to avoid cross-links between unlabeled and ¹⁵N-labeled peptides. We have revised the description of the preparation method and copied below. The description of the preparation method for the E.coli-SS-¹⁵N dataset was correct, since disulfide bonds were endogenous cross-links, there was no cross-linking reaction in sample preparation.

Preparation of the E.coli-Leiker-¹⁵N dataset

Leiker bAL2 was used to cross-link the *E. coli* MG1655 whole-cell lysates. The CXMS experiment was conducted as previously described²². MG1655 was cultured in unlabeled or ¹⁵N-labeled M9 medium and harvested at OD₆₀₀ 0.6-0.8. Pellets of 40 OD*ml *E. coli* cells was resuspended in 0.4 ml lysis buffer (50 mM HEPES, pH 7.5, 150 mM NaCl). Cell lysates were prepared using a FastPrep homogenizer (6.5 m/s, 20 s, repeat 5 times). After measuring protein concentration, 1 mg of unlabeled and ¹⁵N-labeled lysate was cross-linked separately with 0.33 mg [d0]-bAL2 for 0.5 hour at room temperature. Cross-linking reactions were quenched with 20 mM ammonium bicarbonate. The two reactions were mixed together and then precipitated by TCA followed by Trypsin digestion. After filtering with a 50-kD cutoff Amicon Ultra-0.5 Centrifugal Filter Unit, the digested peptides were brought to a volume of 3 mL with 2% ACN, 20 mM HEPES, pH 8.2; the pH was adjusted to 10.0 with ammonia. High-pH reverse phase separation was used for fractionation. The peptides were eluted with buffer B (80% ACN, 5 mM NH₄COOH, pH 10) gradient. A total of 39 two-min fractions were collected, and then combined into five fractions of similar shades of color (bAL2-linked peptides are bright yellow before cleavage of the biotin tag). Each pooled sample was evaporated to 200–300 µl before enrichment of bAL2-linked peptides on 50 µl high-capacity streptavidin beads. After release off beads and desalting through a C18 column, the five fractions of bAL2-linked peptides were analysed by LC-MS/MS using a Q Exactive HF mass spectrometer coupled with an EASY-nLC 1,000 system (both from Thermo Fisher Scientific). Precursors of the +1, +2, +8 or above, or unassigned charge states were rejected and dynamic exclusion was set to 20 s.

Minor points:

1. I would like the authors to change the question sentences on Page 3 lines 80-83 to statement sentences.

Reply 1-5:

Thank you for your advice. We have turned the questions into statements (lines 80-83 on page 3 in the revised main text, copied below).

Initial version:

Therefore, TDA becomes the only general method for credibility evaluation, but is

it sufficient? While all search engines seem to have justified their existence by TDA, can we trust them equally well in cross-link identification, especially at a proteome scale?

Revised version:

Therefore, TDA becomes the only general method for credibility evaluation, but it has not been validated at a proteome scale, and whether or not to separately control the TDA-FDR for inter-protein and intra-protein cross-links is still controversial.

2. Please rephrase the following sentence on page 8 lines 256-257.

Reply 1-6:

Thank you for your advice. We have added the references and explained the recall rate of the coarse-scoring algorithm (lines 259-264 on page 8 in the revised main text, copied below). Hope we understand what you mean.

Initial version:

However, thanks to the good fragmentation of α -peptides and the strong-performance of the coarse-scoring algorithm, pLink 2 recalled 98.8% of correct α -peptides in top-5, even with the huge human database (Fig. 2c).

Revised version:

However, thanks to sufficient fragmentation of α -peptides in most cases^{12, 15} and the well-designed fragment index (see “Fragment index based α -peptide retrieval” in Methods), the recall rate of the coarse-scoring algorithm of pLink 2 was as high as 98.8%, *i.e.* it was able to retain the correct α -peptide sequence among the top-5 highest scoring candidate sequences 98.8% of the time, even when searching against the human database (Fig. 2c).

3. Page 12, lines 383-385, “Despite this, consider that many studies have shown that the FDR of intersection results drops rapidly”. Please rephrase this sentence.

Reply 1-7:

Thank you for your advice. We have rephrased the sentence (lines 403-404 on page 12 in the revised manuscript, copied below).

Initial version:

Despite this, consider that many studies have shown that the FDR of intersection results drops rapidly⁴⁹⁻⁵¹.

Revised version:

Nevertheless, many studies have shown that the credibility of intersection results is greatly enhanced⁴⁹⁻⁵¹.

Reviewer #2 (Remarks to the Author):

In their paper entitled “pLink 2: A high-speed search engine with systematic evaluation for proteome-scale identification of cross-linked peptides”, Chen et al. propose a new software tool for the confident identification of cross-linked peptides in complex mixtures analyzed by high resolution mass spectrometry. They benchmark their new tool, in particular against pLink1 and Kojak using different types of datasets and show that pLink2 overperforms in speed, precision and sensitivity.

Reply 2-1:

Thank you very much for reviewing our paper.

Even if the paper looks interesting, there are major issues that precludes its publication in *Nature Communications*. The first and major one relies in the fact that this is a very technical paper, with a lot of details on the informatics behind the software and *Nature Communications* does not appear as the best journal to publish such results. A more bioinformatics-centric journal such as *Bioinformatics* would be a better place for such paper.

Reply 2-2:

Thanks for your comments. In our paper, we have evaluated the performance of ten search engines (Table 1 and Supplementary Note 1), and the evaluation results show that currently available software tools for CXMS data analysis suffered from poor speed and credibility, and pLink 2 achieved the highest speed, precision and sensitivity by a large margin at proteome scales. Although there were some informatics details in our paper, we have demonstrated the necessity of these informatics technologies (Supplementary Note 6), and we want to emphasise that it was these informatics technologies that made pLink 2 far superior to others. We hope that by spreading through the high-profile journal *Nature Communications*, the concept of systematic evaluation of software tools (FDR is one but not all) will be better appreciated as it should be. This concept and the informatics technologies devised in this work will greatly improve future development of MS search engines.

A second issue relies on the fact that the datasets that were used are not described, do not represent a real sample, and thus there may be a bias in favor of pLink2 using such datasets.

Reply 2-3:

A total of twelve datasets were used in our paper, which were listed in Supplementary Table 2 (copied below) and were described in detail in the Datasets section of the Methods section (copied below). Eight of twelve datasets represented real samples and were obtained from previous published studies, and four of twelve datasets were prepared in our paper.

Among four new datasets, two were simulated datasets and two were ^{15}N metabolically labeled datasets. For the simulated datasets, although they do not represent real samples, their fragmentation characteristics were adapted from two synthetic datasets, which were two real samples. For the ^{15}N metabolically labeled datasets, they were two *E. coli* real samples.

In order to guarantee the fairness of four new datasets, the method used to generate two simulated datasets was described in detail in Supplementary Note 8 and the source code was published in GitHub (<https://github.com/pFindStudio/pLink2/tree/master/pSimXL>), and two ^{15}N metabolically labeled datasets were uploaded to ProteomeXchange with Project accession: PXD012109, Username: reviewer21162@ebi.ac.uk, and Password: puL7ioYP.

In summary, all twelve datasets used in our paper were described in detail, with guaranteed fairness for all search engines.

Supplementary Table 2. Detailed information of the twelve datasets used in this study

Dataset	Cross-linker	Mass spectrometer	# Files (RAW / MGF)	# MS2 scans	Reference
Simulated-BS3	BS3	-	1	10,000	-
Simulated-SS	SS	-	1	10,000	-
Synthetic-BS3	BS3	LTQ-Orbitrap-ETD	1	2,077	[1]
Synthetic-SS	SS	LTQ-Orbitrap-ETD	1	5,000	[16]
E.coli-Leiker- ^{15}N	Leiker	Q Exactive HF	5	258,555	-
E.coli-SS- ^{15}N	SS	Q Exactive HF	10	289,381	-
SCF(FBXL3)-BS3	BS3	Q Exactive	3	81,300	[14]
Ca _v 1.1-SS	SS	Q Exactive	2	95,339	[25]
E.coli-Leiker	Leiker	Q Exactive	116	2,144,734	[15]
C.elegans-Leiker	Leiker	Q Exactive	36	687,916	[15]
E.coli-SS	SS	Q Exactive	98	2,848,512	[16]
Human-SS	SS	Q Exactive	32	966,351	[16]

Datasets

A total of twelve datasets were used to evaluate the credibility of pLink 2. They can be categorized into two classes. The first class contained eight datasets serving for demonstration of four new evaluation methods. Specifically, they were Simulated-BS3 and Simulated-SS for the simulated dataset evaluation, Synthetic-BS3 and Synthetic-SS for the synthetic dataset evaluation, E.coli-Leiker- ^{15}N and E.coli-SS- ^{15}N for the ^{15}N metabolically labeled dataset evaluation, and SCF(FBXL3)-BS3 and Ca_v1.1-SS for the entrapment database evaluation. These datasets were used to systematically evaluate the performance of different search engines, *i.e.*, the speed, sensitivity, and precision, independent of common evaluation methods such as the TDA-FDR method and crystal structures.

The second class contained four previously published proteome-scale datasets

already analysed by pLink 1, including E.coli-Leiker, C.elegans-Leiker, E.coli-SS, and Human-SS, and they were reanalysed to demonstrate the performance and versatility of pLink 2. The detailed information of these twelve datasets are shown in Supplementary Table 2.

Preparation of two simulated datasets

The fragmentation characteristics of synthetic peptides cross-linked by the BS3¹⁰ and disulfide bonds¹³ were used to generate Simulated-BS3 and Simulated-SS respectively. The Simulated-BS3 dataset consists of cross-linked, loop-linked, mono-linked, and regular MS2 spectra, 2,500 for each type, resulting in 10,000 MS2 spectra in total. As there are no mono-linked peptides in a disulfide bond sample, the Simulated-SS dataset only consists of 2,500, 2,500, and 5,000 for cross-linked, loop-linked, and regular MS2 spectra, respectively, resulting in 10,000 MS2 spectra in total. The simulation method is described in Supplementary Note 8.

Preparation of the Synthetic-BS3 dataset

The Synthetic-BS3 dataset was a collection of 2,077 annotated high-energy collisional dissociation (HCD) spectra obtained from 38 synthetic peptides cross-linked pairwise through BS3¹⁰. Of the 2,077 spectra, 1,030 were from light [d0]-BS3 and 1,047 were from heavy [d4]-BS3. The 1,047 spectra of cross-linked peptides containing heavy [d4]-BS3 served as negative samples because in this study we searched with the mass of light [d0]-BS3 and the precursor mass tolerance was as small as ± 20 ppm. The Synthetic-BS3 dataset was prepared as described previously¹⁰.

Preparation of the Synthetic-SS dataset

The Synthetic-SS dataset was a collection of 5,000 annotated HCD spectra obtained from 72 cysteine-containing synthetic peptides cross-linked pairwise through disulfide bonds¹³. Of the 5,000 spectra, 2,289 were high-quality HCD spectra of disulfide-linked peptide pairs obtained from 72 synthetic peptides and 2,711 were negative samples that were not identified as regular peptides or disulfide-linked peptide pairs. The Synthetic-SS dataset was prepared as described previously¹³.

Preparation of the E.coli-Leiker-¹⁵N dataset

Leiker bAL2 was used to cross-link the *E. coli* MG1655 whole-cell lysates. The CXMS experiment was conducted as previously described²². MG1655 was cultured in unlabeled or ¹⁵N-labeled M9 medium and harvested at OD₆₀₀ 0.6-0.8. Pellets of 40 OD*ml *E. coli* cells was resuspended in 0.4 ml lysis buffer (50 mM HEPES, pH 7.5, 150 mM NaCl). Cell lysates were prepared using a FastPrep homogenizer (6.5 m/s, 20 s, repeat 5 times). After measuring protein concentration, 1 mg of unlabeled and ¹⁵N-labeled lysate was cross-linked separately with 0.33 mg [d0]-bAL2 for 0.5 hour at room temperature. Cross-linking reactions were quenched with 20 mM ammonium bicarbonate. The two reactions were mixed together and then precipitated by TCA followed by Trypsin digestion. After filtering with a 50-kD cutoff Amicon Ultra-0.5 Centrifugal Filter Unit, the digested peptides were brought to a volume of 3 mL with 2% ACN, 20 mM HEPES, pH 8.2; the pH was adjusted to 10.0 with ammonia. High-pH reverse phase separation was used for fractionation. The peptides were eluted with buffer B (80% ACN, 5 mM

NH₄COOH, pH 10) gradient. A total of 39 two-min fractions were collected, and then combined into five fractions of similar shades of color (bAL2-linked peptides are bright yellow before cleavage of the biotin tag). Each pooled sample was evaporated to 200–300 µl before enrichment of bAL2-linked peptides on 50 µl high-capacity streptavidin beads. After release off beads and desalting through a C18 column, the five fractions of bAL2-linked peptides were analysed by LC-MS/MS using a Q Exactive HF mass spectrometer coupled with an EASY-nLC 1,000 system (both from Thermo Fisher Scientific). Precursors of the +1, +2, +8 or above, or unassigned charge states were rejected and dynamic exclusion was set to 20 s.

Preparation of the E.coli-SS-¹⁵N dataset

The *E. coli* strain BL21 was cultured in unlabeled and ¹⁵N-labeled NH₄Cl (Cambridge Isotope Laboratories, Inc.) M9 medium separately and harvested at OD₆₀₀=0.75. From a mixture of 35 OD*mL unlabeled and 35 OD*mL ¹⁵N-labeled cells, periplasmic proteins were prepared using the osmotic shock method as described before¹³. Periplasmic proteins were released into 6 mL of pre-cooled solution of 5 mM MgSO₄ supplemented with 2 mM NEM, and then precipitated on ice with 25% trichloroacetic acid (TCA) followed by cold acetone wash twice. Precipitated proteins were air dried, resuspended in 8 M urea, 100 mM Tris, 2 mM NEM, pH 6.5. After the protein concentration was measured using the BCA Protein Assay Kit (Pierce), the sample was brought to 1–2 mg/mL, digested sequentially with Lys-C, trypsin and Glu-C before SCX fractionation as described in our step-by-step protocol⁶⁰. Eight SCX fractions were collected by sequential elution with 35 µl of 50, 150, 250, 350, 500, 650, 800, 1M ammonium acetate, pH 2–3, at a flow rate of 1.0–2.0 µl/min. The LC-MS/MS analysis was performed on a Q-Exactive HF mass spectrometer coupled to an Easy-nLC 1,000 II system (Thermo Fisher Scientific). Peptides were loaded on a pre-column (75 µm ID, 6 cm long, packed with ODS-AQ 120 Å–10 µm beads from YMC Co., Ltd.) and further separated on an analytical column (75 µm ID, 14 cm long, packed with C18 1.9 µm 100 Å resin from Welch Materials) with a linear reverse-phase gradient from 100% buffer A (0.1% formic acid in H₂O) to 28% buffer B (0.1% formic acid in acetonitrile) in 56 (or 71) min at a flow rate of 250 nl/min. The top-12 most intense precursor ions from each full scan (resolution 60,000) were isolated for HCD MS2 (resolution 15,000; normalized collision energy 27) with a dynamic exclusion time of 40 s. Precursors with 3+ to 6+ charge states were included. Each sample was analysed a second time using the same parameters except that 2+ precursors were also included for HCD MS2 to find more disulfide bonds in loop-linked peptides.

Preparation of two datasets used in the entrapment database method

The RAW files of SCF(FBXL3)-BS3 dataset¹⁴ and Ca_v1.1-SS dataset⁴⁸ were kindly provided by the authors of [14] and [48], respectively. Two complexes were expressed and purified as described therein.

Preparation of four cell lysate datasets

The RAW files of E.coli-SS and Human-SS datasets cross-linked by disulfide bonds¹³, and E.coli-Leiker and C.elegans-Leiker datasets cross-linked by Leiker²² were kindly provided by the authors of [13] and [22], respectively. They were prepared as described therein.

Finally, the first part of the software is similar to another one, which is not mentioned here which is MassSpecStudio 2.0 that works also well for entire proteomes. This is clearly not the case for all software tools dedicated to proteome-wide analysis of cross-linked data and thus a comparison with this particular tool would be very useful.

Reply 2-4:

Thanks for bringing MassSpecStudio 2 (*Sarpe et al., Mol. Cell. Proteomics 2016*) to our attention. In the revision, we compared pLink 2 and MassSpecStudio 2 in both design and performance.

Firstly, we want to point out three major differences between pLink 2 and MassSpecStudio 2, which highlights the novelties of pLink 2.

1. **The search strategy.** MassSpecStudio 2 employs a peptide library reduction strategy by pre-scoring of linear peptides to reduce the search space of cross-linked peptides (Fig. 2 of the MassSpecStudio 2 paper, copied below). Its peptide library reduction strategy was similar to the open search strategy employed in pLink 1 (*Yang et al., Nat. Methods 2012*), Protein Prospector (*Trnka et al., Mol. Cell. Proteomics 2014*), and Kojak (*Hoopmann et al., J. Proteome Res. 2015*), which looks for candidates of α -peptide and β -peptide simultaneously and then tries to combine the candidates properly.

As the fragmentation efficiency of α -peptide is usually better than that of the β -peptide, α -peptide typically has a better pre-score than β -peptide. Due to the

different but over-lapping pre-score distributions of α - and β -peptides, it is difficult to decide what pre-score threshold to use in MassSpecStudio 2; a strict threshold may lose the correct β -peptide, and a loose threshold may greatly impede peptide library reduction.

In contrast, to exploit the unequal fragmentation efficiencies of α -peptide and β -peptide, pLink 2 adopted a two-stage open search strategy (Fig. 1 of the pLink 2 manuscript, copied below): for the α -peptide, only the top-5 coarse-scored candidates were kept; then for each candidate of the α -peptide, the mass of the β -peptide can be deduced and used to search all the candidates for the β -peptide, whose number is usually small under high-accuracy mass spectrometry. This two-stage open search strategy of pLink 2 combined with the index structure (see below the second difference) not only enables effective screening of the α -peptides, but also prevents the loss of the correct β -peptides in the stage of looking for candidate sequences.

Fig. 1. Workflow of pLink 2. **a)** The general workflow. Step 1, MS1 scans are preprocessed by pParse to extract precursor candidates. Step 2, for each MS2 spectrum, α -peptide candidates are retrieved from the fragment index using query peaks generated from the spectrum. Step 3, β -peptide candidates are retrieved from the peptide index using the complementary masses of α -peptides. Step 4, α - and β -peptide candidates are paired and fine-scored with the MS2 spectrum. Step 5, all top scored PSMs are reranked and filtered after FDR control. **b)** The sub-workflow of α -peptide retrieval. For each MS2 spectrum, the peaks are converted into regular b , y ions to query the fragment index. Only those peptides with at least two matched ions are coarse-scored with the spectrum, and the top-5 coarse-scored α -peptide candidates are kept. **c)** The sub-workflow of β -peptide retrieval. For each α -peptide candidate, the open mass is first calculated by subtracting the α -peptide mass and the cross-linker mass from the precursor mass, and this mass is used to retrieve β -peptide candidates from the peptide index. Then, each of the five α -peptide candidates is paired with each of its complementary β -peptide candidates and these pairs are fine-scored with the

spectrum. Finally, the highest fine-scored peptide pair is kept. **d)** The reranking algorithm. PSMs are grouped into intra-protein, inter-protein, loop-linked, mono-linked, and regular groups, and a semi-supervised learning algorithm is used to rescore and rerank them in each group.

- The index structure.** Actually, the idea of open search strategy had long existed since Singh et al., Anal. Chem. 2008, but most open search engines are not as efficient as pLink 2 (Table S1 in Supplementary Note 1). The key factor of the high efficiency of pLink 2 is the fragment index. With the fragment index, α -peptide candidates with matched peaks are efficiently retrieved and coarse-scored, reducing the number of coarse-scored α -peptides to less than 1% of those without the fragment index (Supplementary Note 6), thus greatly accelerating pLink 2. In contrast, the paper of MassSpecStudio 2 did not mention any index structure, and did not compare the computing time of MassSpecStudio 2 with other search engines. In our performance evaluation, pLink 2 was more than 200 times faster than MassSpecStudio 2, showing the high efficiency of pLink 2 at a proteome scale (Supplementary Note 5).
- The coarse-scoring algorithm.** Although both MassSpecStudio 2 and pLink 2 use a probabilistic coarse-scoring algorithm, their details are different. MassSpecStudio 2 adapted the E-score from OMSSA (Geer et al., J. Proteome Res. 2004), which only measures the number of matched ions. In contrast, pLink 2 adapted the pre-scoring from pLink 1 (Yang et al., Nat. Methods 2012), which measures not only the number of matched ions, but also the average intensity ranking of matched ions and the length of the longest sequence tag. Additionally, MassSpecStudio 2 only considers regular *b* and *y* ions, while pLink 2 also considers *xlink b* and *y* ions by converting them to complementary ions (see “Fragment index based α -peptide retrieval” in the Methods section of pLink 2 manuscript). This comprehensively designed coarse-scoring algorithm coupled with the fragment index enables pLink 2 to retrieve α -peptides with high efficiency and effectiveness. We have demonstrated that the sensitivity for α -peptides in top-5 reached at least 98.6% in Supplementary Note 6.

Secondly, following your advice, we have included MassSpecStudio 2 for comparison, which was placed in Supplementary Note 5 and copied below. The main conclusions were that MassSpecStudio 2 identified markedly fewer intra-protein peptide pairs, not only fewer than those identified by pLink 2, but also fewer than the inter-protein peptide pairs identified by MassSpecStudio 2 itself. This was highly abnormal since intra-protein cross-links are more readily observed than inter-protein cross-links, and most search engines identified more intra-protein cross-links than inter-protein ones. Furthermore, MassSpecStudio 2 took more than 2 weeks to finish searching the E.coli-SS-¹⁵N dataset, while pLink 2 only took 1 hour, hence the low efficiency of MassSpecStudio 2 made it impractical for proteome-scale cross-linked peptide identification.

In summary, pLink 2 is very different from MassSpecStudio 2, in both software design and performance. Supplementary Note 5 shows that pLink 2 outperformed MassSpecStudio 2 in sensitivity, precision, and speed.

Supplementary Note 5. Evaluate the performance of MassSpecStudio 2

MassSpecStudio 2 is another search engine for cross-linked peptide identification¹³, and it employed a pre-scoring on linear peptides to reduce the search space of cross-linked peptide combinations. Its peptide library reduction strategy was similar to the open search strategy that had been widely used in pLink 1¹, Protein Prospector³, and Kojak¹⁴. Similar to XlinkX, MassSpecStudio 2 could not analyse MGF files, and thus could not be evaluated using simulated datasets or synthetic datasets. Furthermore, MassSpecStudio 2 only reported identifications at peptide level, and thus could not be evaluated using entrapment databases like XlinkX in Supplementary Note 4. We finally evaluated the performance of MassSpecStudio 2 using ¹⁵N metabolically labeled datasets. The search parameters for MassSpecStudio 2 were the same as in Supplementary Table 5 except that the FDR threshold was unknown and could not be set in MassSpecStudio 2. For each identified cross-linked peptide pair, MassSpecStudio 2 reported only one “Best MS2 Scan Number”, we thus used that MS2 scan to calculate the ¹⁵N quantitation ratio.

For the E.coli-Leiker-¹⁵N dataset, as MassSpecStudio 2 was very slow, the database for MassSpecStudio 2 search contained only 1,284 proteins identified from the samples using pFind (same as in Supplementary Note 4). pLink 2 also searched against the same restricted database of 1,284 proteins, so that results could be fairly compared. MassSpecStudio 2 identified 242 inter-protein peptide pairs and 201 intra-protein peptide pairs (Fig. S11), and this phenomenon was highly abnormal since intra-protein cross-links are more readily observed than inter-protein cross-links and most search engines identified more intra-proteins than inter-proteins (Fig. 3 in main text). Additionally, both inter-protein and intra-protein peptide pairs uniquely identified by MassSpecStudio 2 had much higher percentage of NaN-ratios compared with those uniquely identified by pLink 2. For the E.coli-SS-¹⁵N dataset, similarly, the database for MassSpecStudio 2 and pLink 2 searches contained only 1,181 proteins identified from the samples using pFind. Evaluation results were similar to those obtained for the E.coli-Leiker-¹⁵N dataset (Fig. S12).

Fig. S11. Compare pLink 2 with MassSpecStudio 2 on the E.coli-Leiker-¹⁵N dataset. At peptide pair level, the Venn diagram for **a**) inter-protein and **b**) intra-protein peptide pairs identified by MassSpecStudio 2 and pLink 2. Numbers in parentheses indicate the number and the percentage of NaN-ratio results that lie in the corresponding region. For example, 234 (38, 16.2%) means that

MassSpecStudio 2 uniquely identified 234 inter-protein peptide pairs, of which 38 were NaN-ratios, accounting for 16.2% of 234.

Fig. S12. Compare pLink 2 with MassSpecStudio 2 on the E.coli-SS-¹⁵N dataset. At peptide pair level, the Venn diagram for **a**) inter-protein and **b**) intra-protein peptide pairs identified by MassSpecStudio 2 and pLink 2.

MassSpecStudio 2 and pLink 2 were also compared in terms of computing time. Where possible, 8 threads were used for each comparison. The computing times for MassSpecStudio 2 and pLink 2 (Windows Server, Intel Xeon E5-2670 CPU with 32 cores, 2.6 GHz, 128 GB RAM) are shown in Fig. S13. With restricted databases, pLink 2 took only 1 hour to finish searching the E.coli-Leiker-¹⁵N dataset or the E.coli-SS-¹⁵N dataset, while MassSpecStudio 2 took more than 200 or more than 300 hours respectively, showing the high efficiency of pLink 2 at a proteome scale.

Fig. S13. The computing times for MassSpecStudio 2 and pLink 2 on the E.coli-Leiker-¹⁵N dataset and the E.coli-SS-¹⁵N dataset. Both MassSpecStudio 2 and pLink 2 searched against restricted databases containing only 1,284 proteins and 1,181 proteins identified by pFind for the E.coli-Leiker-¹⁵N dataset and the E.coli-SS-¹⁵N dataset respectively.

REVIEWERS' COMMENTS:

Reviewer #1 (Remarks to the Author):

In the revised manuscript, the authors have satisfactorily addressed most of the concerns I raised previously therefore I support its publication in Nature Communications.

My last comment is about Xi, one of the softwares used for comparison with pLink2 in this study. The authors state that Xi do not support proteome-scale cross-linked peptide identification but I've seen many times in conferences that Xi supports proteome-wide crosslinking studies. However, I couldn't find any publications in peer-reviewed journals describing Xi (except for a link to download the software from github) so I guess it was either not yet published or it's only for internal use. For clarity, I would like to ask the authors to specify where they obtain the software, which version they use, and possibly provide some identity number to avoid confusion of similar studies in the future.

Reviewer #2 (Remarks to the Author):

The authors have responded to all reviewers' comments, and the manuscript has largely improved in its revised version.

It is therefore now suitable for publication in Nature Communications.

Response to Reviewers' comments

We thank the reviewers for their insightful and very constructive comments, and have revised our manuscript accordingly. We are also grateful to the editor for giving us a chance to submit a revision. Our replies to specific comments and suggestions are as follows.

REVIEWERS' COMMENTS:

Reviewer #1 (Remarks to the Author):

In the revised manuscript, the authors have satisfactorily addressed most of the concerns I raised previously therefore I support its publication in Nature Communications.

Reply 1-1:

Thank you very much for your positive comments.

My last comment is about Xi, one of the softwares used for comparison with pLink2 in this study. The authors state that Xi do not support proteome-scale cross-linked peptide identification but I've seen many times in conferences that Xi supports proteome-wide crosslinking studies. However, I couldn't find any publications in peer-reviewed journals describing Xi (except for a link to download the software from github) so I guess it was either not yet published or it's only for internal use. For clarity, I would like to ask the authors to specify where they obtain the software, which version they use, and possibly provide some identity number to avoid confusion of similar studies in the future.

Reply 1-2:

Thank you for your advice. To the best of our knowledge, Xi was first published in a peer-reviewed journal (Giese et al., Mol. Cell. Proteomics 2016), and then was described on a preprint server (Mendes et al., bioRxiv 2019). The MCP paper of Xi explored the detailed fragmentation behavior of cross-linked peptides in CID mode, and then proposed a search strategy Xi to identify cross-linked peptides using the fragmentation behavior explored previously. The bioRxiv paper of Xi introduced sequential digestion for cross-linking MS analysis, and then reported the software Xi entirely. From the descriptions of Xi in two papers, we think that the algorithms of Xi in two papers were the same. The bioRxiv paper confirmed this by stating that "The algorithm of Xi has been described conceptually before³³" in which the 33rd reference was the MCP paper of Xi.

According to our tests, Xi supports cross-link identification against hundreds of proteins (Table 1 in the main text), but it has not been shown to support cross-link identification against the *full human proteome* database (Supplementary Table 8 in Supplementary Note 1). Nevertheless, Xi can be used for proteome-wide searches when the database has been pre-filtered based on standard bottom-up proteomics. The

bioRxiv paper of Xi described that a restricted database according to the iBAQ value was used for proteome-wide cross-linking analysis (Data analysis section of the bioRxiv paper of Xi and copied below)

Data analysis section of the bioRxiv paper of Xi

“... for the 26S proteasome a linear search of the sample was first performed using a complete *S. cerevisiae* database and the 1% most intense proteins accordingly to the iBAQ value were selected to build the database for the crosslinking search; for each one of the cytosolic SEC fractions the procedure was as for the 26S proteasome, using as database for the linear searched the complete human database...”

The software Xi used in this manuscript was downloaded from GitHub <https://github.com/Rappsilber-Laboratory/XiSearch>, on May 15, 2018 (version 1.6.731). For clarity, we have added the download links of all software tools used in this study and cited both the MCP paper and the bioRxiv paper of Xi (Supplementary Table 1 and copied below). Furthermore, we also indicated that Xi can be used for proteome-wide searches when using a restricted database (last paragraph of Supplementary Note 1 and copied below).

Supplementary Table 1. The ten search engines used in this study

Search Engine	Website	Version	Search Strategy ^a	Publication Year	Reference
xQuest / xProphet	http://proteomics.ethz.ch/cgi-bin/xquest2.cgi/download.cgi	2.1.1	E / O	2008 / 2012	[1, 2]
StavroX	http://www.stavrox.com/Download.htm	3.6.0.1	E	2012	[3]
pLink 1 (pLink / pLink-SS)	http://pfind.ict.ac.cn/software/pLink1/index.html	1.23	E / O	2012 / 2015	[4, 5]
Protein Prospector	http://prospector.ucsf.edu/prospector/mshome.htm	v5.21.2	O	2014	[6]
Kojak	http://www.kojak-ms.org/download.html	1.5.5	O	2015	[7]
Xi	https://github.com/Rappsilber-Laboratory/XiSearch	1.6.731	O	2016 / 2019	[8, 9]
Xilmass	https://github.com/compomics/xilmass	1.0	E	2016	[10]
MetaMorpheusXL	https://github.com/smith-chem-wisc/MetaMorpheus/releases	0.0.285	O	2018	[11]
Xolik	http://bioinformatics.ust.hk/Xolik.html	0.3	E	2018	[12]
pLink 2	http://pfind.ict.ac.cn/software/pLink/index.html	2.2	O	-	[13]

^a E for exhaustive search and O for open search.

Last paragraph of Supplementary Note 1

To the best of our knowledge, xQuest, Xilmass, StavroX, and Xi have not been shown to support cross-linked peptide identification against the *full human proteome* database, while the other six do. Some search engines in the former group may be used for proteome-wide searches assisted by additional techniques. For example, xQuest used an isotopically labeled cross-linker to search against the full *E. coli* proteome¹, and Xi used a restricted database based on standard bottom-up proteomics to search against the human proteome⁹. In contrast, search engines in the latter group can be used for proteome-wide searches without additional techniques. They can be further categorized into three types according to sensitivity: Xolik and MetaMorpheusXL have very low sensitivities; PP and Kojak

have moderately higher sensitivities; pLink 1 and pLink 2 have the highest sensitivities. Performance comparison among all ten search engines using synthetic datasets shows that pLink 2, pLink 1, and Kojak were indeed the top-3 highest-sensitivity search engines, which confirmed the conclusion drawn from the evaluation using simulated datasets.

Reviewer #2 (Remarks to the Author):

The authors have responded to all reviewers comments, and the manuscript has largely improved in its revised version.
It is therefore now suitable for publication in Nature Communications.

Reply 2-1:

Thank you very much for your positive comments.